



# Can explicit convection improve modelled dust in summertime West Africa?

Alexander J Roberts[1], Margaret J Woodage[2], John H. Marsham[1,3], Ellie J. Highwood[2], Claire L. Ryder[2], Willie McGinty[4], Simon Wilson[4], Julia Crook[1]

[1]School of Earth and Environment, University of Leeds, LS2 9JT, Leeds, UK

[2]Department of Meteorology, University of Reading, RG6 6BB, Reading UK

[3]National Centre for Atmospheric Science, University of Leeds, Leeds, UK

[4]NCAS-CMS, Department of Meteorology, University of Reading, Reading, UK.

*Correspondence to*:a.j.roberts1@leeds.ac.uk

**Abstract.** Global and regional models have large systematic errors in their modelled dust fields over West Africa. It is well established that cold pool outflows from moist convection (haboobs) can raise over 50 % of the dust over the Sahara and Sahel in summer, but parameterised moist convection tends to give a very poor representation of this in models. Here, we test the hypothesis that an explicit representation of convection improves haboob winds and so may reduce errors in modelled dust

fields. The results show that despite varying both grid-spacing and the representation of convection there are only minor changes in dust aerosol optical depth (AOD) and dust mass loading fields between simulations. In all simulations there is an AOD deficit over the observed central Saharan dust maximum and a high bias in AOD along the west coast: both features consistent with many climate (CMIP5) models. Cold pool outflows are present in the explicit simulations and do raise dust. Consistent with this there is an improved diurnal cycle in dust-generating winds with a seasonal peak in evening winds at

locations with moist convection that is absent in simulations with parameterised convection. However, the explicit convection does not change the AOD field significantly for several reasons. Firstly, the increased windiness in the evening from haboobs is approximately balanced by a reduction in morning winds associated with the breakdown of the nocturnal low-level jet (LLJ). Secondly, although explicit convection increases the frequency of the strongest winds, these are still weaker than observed, especially close to the observed summertime Saharan dust maximum: this results from the fact that although large mesoscale

convective systems (and resultant cold pools) are generated, they have a lower frequency than observed and haboob winds are too weak. Finally, major impacts of the haboobs on winds occur over the Sahel, where, although dust uplift is known to occur in reality, uplift in the simulations is limited by a seasonally constant bare soil fraction in the model, together with soil moisture and clay fractions which are too restrictive of dust emission in seasonally-varying vegetated regions. For future studies, the results demonstrate 1) the improvements in behaviour produced by the explicit representation of convection, 2) the value of

simultaneously evaluating both dust and winds and 3) the need to develop parameterisations of the land surface alongside those of dust-generating winds.



## 1 Introduction

During the summer season the Sahara is the world's largest source of mineral dust (Ginoux et al., 2012, Prospero et al., 2002) and representations of dust are known to improve numerical weather prediction (NWP) models (Haywood *et al.*, 2005; Tompkins et al., 2005; Rodwell and Jung, 2008), although dust forecasts skill remains limited (Chaboureau et al., 2016, Huneeus et al., 2016, Terradellas et al., 2016). Dust is also a prognostic variable in several climate models, although its value has been questioned due to the poor performance of the models in representing dust variability (Evan et al., 2014). There is, therefore, a need to improve dust models across time scales, and a need to improve the representation of both the land surface that emits dust and dust-generating winds. For winds it is known that rare, high-wind speed events are disproportionately important for the raising of dust (Cowie et al., 2015) and that a poor representation of cold-pool outflows from moist convection (haboobs: Roberts and Knippertz, 2012) is one major limitation of summertime winds in current models for the Sahara and Sahel (Marsham et al 2011; Knippertz & Todd 2010). Haboobs can range in size from 10s to 100s of km across and rare, large events can be some of the largest single uplift events in West Africa (Roberts and Knippertz 2014). Although often considered a Sahelian phenomenon (in West Africa), haboobs were shown by Marsham et al. (2013) and Allen et al. (2013) to be observed commonly at Bordj Badji Mokhtar in the central Sahara (21.38°N, 0.92°E) during June of 2011. Rainfall retrievals (Tropical Rainfall Measuring Mission) also indicate that precipitating clouds are present north of the position of the analysed intertropical discontinuity (as much as 5 degrees) at times of monsoon surges (Figure 9 in Roberts et al., 2015). This is important, not because of the likelihood of rainfall reaching the surface, but because (consistent with Marsham et al., 2013; Allen et al., 2013; Trzeciak et al., 2017), cold pools/haboobs can be generated north of the analysed monsoon flow in a region with a deep dry boundary layer and a deflatable surface soil (the Sahara).

Several meteorological processes are known to raise mineral dust. Synoptic-scale systems (Johnson and Osborne, 2011) and the breakdown of nocturnal low-level jets (Knippertz, 2008, Fiedler et al., 2013) are of sufficiently large scale to be captured by many models (Woodage et al., 2010, Johnson et al., 2011). However, it is estimated that dust raised by convectively generated cold-pool outflows contribute over 50% of the total uplift in some areas of the Sahara in summer (Marsham et al., 2013; Allen et al., 2013 & 2014; Heinold et al., 2013) and may explain the seasonal cycle of dust in the Sahel and Sahara (Marsham et al. 2008). The parameterised representation of convection in global models can make haboobs essentially non-existent (Marsham et al., 2011) and consistent with this, data assimilation has shown that an NWP model with prognostic dust underestimates dust in regions of observed haboobs (Pope et al. 2016). Comparison of observed near surface winds with meteorological reanalyses in key dust uplift areas (Largeron et al., 2015, Roberts et al., 2017) highlights that even such analyses which are constrained by assimilation of available observations (and often used as de-facto observations) have large systematic biases. In particular, the distribution of wind-speeds in analyses misses the high wind-speed tail, the seasonal and diurnal cycles have amplitudes that are too small and the seasonal evening peak in winds associated with cold pools is missing.

A common feature of many previously conducted evaluations of models or analyses is that they evaluate only the dust (usually AOD, e.g. Johnson 2011; Pérez et al., 2011, Chaboureau et al., 2016) or the winds (e.g. Largeron et al., 2015,



Roberts et al., 2017) and not both the dust emission and surface winds. This is despite it being known that there are likely to be systematic biases in both model winds and dust. Without an investigation of the winds alongside the dust it is impossible to judge whether a successful replication of dust fields are as a result of compensating errors, or whether all process involved (including transport and deposition) are correctly represented.

5   Recent modelling work has attempted to address the role of haboobs in models by resolving convection explicitly with high-resolution simulations (Cascade; Birch et al., 2014, Pearson et al, 2014) and applying an offline dust model (Heinold et al., 2013); this highlighted the importance of convective cold pools as well as the representation of near-surface night-time stability. Another approach has been the development and application of a haboob parameterisation, in which additional low-level winds are added that are linked to mass fluxes from the convection scheme (Pantillon et al., 2015; 2016). This approach 10 led to an improved agreement between the potential dust uplift in convection permitting simulations and those with parameterised convection. However, this method obviously does not seek to correct the diurnal cycle bias in rainfall (where peak rain occurs close to midday in parameterised convection simulations, and in the evening in convection permitting simulations and in reality), or evaluate in any detail winds from convection permitting simulations against observations. Chaboureau et al. (2016) compared in-line simulations with both explicit and parameterised convection as well as prognostic 15 dust. They show some success in increasing the occurrence of strong winds in the evening (haboobs) when explicitly representing convection, and in improving the dust AOD biases relative to observations by increasing AOD values in the southern Sahara and northern Sahel. They also show improvements to the meridional AOD gradient in the west of the Sahara. However, the variability in AOD at specific sites, including very high values associated with convectively active African Easterly waves, is still under represented even with explicit convection. In Chaboureau et al., (2016) simulations were re-20 initialised daily, preventing the modification of the large-scale monsoon flow by convective storms (Marsham et al 2011; 2013; Garcia Carreras et al 2013). They also encompassed only part of the summer season (25th July - 2nd September 2006; Heinold et al., 2013, 1 June – 30 July 2006; Pantillon et al., 2015 and 1st - 30th June 2011; Chaboureau et al., 2016) so do not show the full seasonal evolution, and were not able to demonstrate clearly the impact of resolved versus parameterised convection in models that were otherwise identical.

25   The Saharan-West African Monsoon Multi-scale Analysis (SWAMMA) project simulations used in this study have a range of horizontal grid-spacing (4km - 40km), and have both convection permitting and parameterised convection setups. They are performed through a full summer season ($1^{st}$ May – $30^{th}$ September 2011) with a fully interactive mineral dust scheme. Although lateral boundary conditions are updated hourly, the size of the domain and duration of the runs means that away from boundaries model fields can diverge from the parent model, allowing the evolution of the hydrological and dust cycles 30 in each simulation. The authors believe this to be the first reported study of large domain multi-day convection-permitting simulations with prognostic dust over the Sahara and Sahel. The approach of using both dust AOD retrievals and observations of near-surface wind speed to evaluate simulations also makes this work novel, and gives an unprecedented opportunity to attribute errors in dust uplift as well as in AOD magnitudes and distributions. The arrangement of this paper is as follows: section 2 describes the model setup, experiments performed, and observations used to validate the model. Results are presented



in section 3 in which model dust AODs, emissions, low-level winds and storm development are compared between the different models and with observations. Discussion of the results and conclusions follow in section 4.

## 2 Data and Methods

### 2.1 Model Setup

SWAMMA simulations use a limited-area version of the UK Met Office (UKMO) Unified Model (UM), based on the HadGEM3-RA regional climate model previously tested at various resolutions over Africa (Moufouma-Okia and Jones, 2015). The UM is designed to function across a wide range of spatial and temporal scales and is used for meteorology and climate research as well as operational numerical weather prediction. The UM (Version 8.2 is used here) consists of a dynamical core (Davies et al., 2005; Staniforth et al., 2006) which describes evolution of the atmosphere as a non-hydrostatic, fully-
compressible fluid. Model levels are terrain following close to the surface but relax to smooth, parallel levels at height. The model has a fixed Eulerian grid but utilises the semi-implicit, semi-Lagrangian time-stepping to advect variables (allowing for mass conservation). Physics packages include a 2-stream radiation code (Edwards et al., 2012), the Joint UK Land Environment Simulator (JULES) land surface exchange scheme (Best 2005, Best et al., 2011), boundary-layer turbulence (Lock and Edwards 2012), cloud microphysics (Wilkinson 2012), and convection (Stratton et al., 2009). The SWAMMA simulations use
a limited area setup with a domain encompassing all of West Africa (approximately 0-35°N and 23°W-35°E). Simulations are conducted at horizontal grid-spacings of 4km, 12km and 40km, all having 70 levels in the vertical. The 12 km and 40 km models have a rigid model lid at a height of 80 km, while the 4 km version has a rigid lid at 40 km height. Model levels are concentrated in the lower atmosphere to better represent meteorological processes. The lateral boundary conditions (horizontal winds and potential temperature) are updated every hour and produced by performing global simulations using the UM on an
N216 (~ 60km) grid (also version 8.2). Global simulations are initialised every 6 hours using European Centre for Medium-range Weather Forecasts (ECMWF) operational analysis data. Sea surface temperatures for the limited-area SWAMMA runs are updated every 6 hours and obtained by re-gridding ECMWF operational analysis data (as above). The UM configuration used for SWAMMA is not dissimilar to that used by the Cascade simulations (the Cascade simulations are a series of simulations over West Africa for a 40 day period in summer 2006, it used different grid-spacing nested into one another and
both parameterised and explicit convection, for further details see Birch et al., 2014, Pearson et al, 2014), and settings in many of the model physics sections, notably the representation of convection, were adopted from there. The SWAMMA simulations are longer than those in Cascade, running from initialisation on 1 May through to 30 September 2011 (153 days); this spans an entire monsoon season, allowing for investigation of the development of the West African Monsoon (WAM) in an unprecedented way.

30          Another important advance from the Cascade work is the inclusion of prognostic interactive dust in the SWAMMA simulations, which allows for the investigation of the dust-raising and transportation characteristics of the model under varying



resolutions and convection options, as well as assessing the radiative impact that dust has on the WAM system. The dust scheme used is that within the Coupled Large-scale Aerosol Simulator for Studies in Climate (CLASSIC; Johnson et al., 2011) scheme, in which dust particles are assumed to be spherical and transported in the atmosphere as six independent tracers undergoing dry deposition through turbulent mixing and gravitational settling as well as wet deposition through washout from

precipitation. Dust emissions are calculated during each model time-step using prognostic model fields. The dust emission scheme utilises the widely-used algorithm of Marticorena and Bergametti (1995) to calculate horizontal flux in each of 9 bins with boundaries at 0.0316, 0.1, 0.316, 1.0, 3.16, 10.0, 31.6, 100., 316 and 1000 μm radius (the largest three of the size modes are only active in saltation processes). Each of the six dust size bins is treated independently by the radiation scheme with spectral properties being calculated from Mie theory. The horizontal dust flux for dust particles in each size bin is calculated

as a function of the cube of the surface friction velocity ($U^*$), the bare soil fraction in the grid-box (shown in Figure 1), the mass fraction of soil particles available at the surface, and a threshold surface friction velocity ($U^*_t$) below which dust is not mobilised. This threshold value is a function of soil moisture in the top layer (10cm thick in the model) and the clay fraction in the grid-box, such that emissions are inhibited for wet soils (further details in Woodward 2011, Ackerley et al., 2012). Dust emission models may be tuned by adjusting coefficients by which $U^*$ and the top level soil moisture are multiplied, together

with a global tuning factor. Here the values used are 1.6, 0.5 and 2.5 respectively, and values were not adjusted for different model grid-spacings in order to make give a fair comparison between the models run at different resolutions. The Harmonized World Soil Database (FAO, 2012) is used to determine soil texture and thus the fractions of clay, silt and sand available in each surface grid-box for the dust emissions scheme. Dust fields are initialised from zero and drop to zero on the lateral boundaries (so that no dust enters the domain at the boundaries). Surface infra-red emissivity is changed from the JULES

default value over bare soil (0.97) to 0.9 for these experiments, as this is more realistic over the Sahara (Ogawa and Schmugge, 2004).

Within the framework described above eight simulations are conducted which comprise the SWAMMA model suite. The main variable factors between the simulations are grid-spacing, representation of convection and radiatively interactive mineral dust (see Table 1). In simulations with parameterised convection the convective scheme in the UM is switched on (Stratton et al.,

2009). This scheme is based on a convective available potential energy (CAPE) closure method, where high CAPE values are identified and tendencies determined to reduce this over a given timescale. In the simulations with explicit convection the convective parameterisation has effectively been switched off by increasing the CAPE closure timescale to a point where CAPE depletion by the parameterisation is insignificant. These models employ a Smagorinsky-style sub grid-scale mixing in all three dimensions (3DS in Table 1) with mixing length constants chosen as those found optimal for the 12km and 4km

models in Cascade (0.05 and 0.1 respectively). In the simulations with radiatively active dust, mineral dust emitted from the surface within the simulations influences the radiation budget via its direct radiative effect (scattering and absorbing solar and thermal radiation); cloud-microphysical effects are not included. While dust is present in the radiatively inactive simulations it does not influence the radiation budget or the evolution of the model meteorology. Comparing simulations with different convection types but dust effects excluded (e.g. 12P and 12E in Table 1) highlights the impact of resolved convection on dust



generation without complications of feedbacks through dust-radiation interactions. We focus on the latter in this paper, although we note here that effects of interactive dust on both dust uplift itself and thermodynamics are far smaller than those of changing the convection (not shown).

## 2.2 Observational data

### 2.2.1 MODIS AOD (TERRA)

We use AOD at 550 nm from the Moderate Resolution Imaging Spectroradiometer (MODIS) Collection 6 'merged' Scientific Data Set (SDS) available from NASA Giovanni online data system (Acker and Leptoukh, 2007). This dataset combines the new 'enhanced Deep Blue' (DB) SDS, now available over all cloud-free and snow-free land surfaces (and therefore including dark vegetated surfaces), and 'dark target' (DT) land and ocean SDS (Sayer et al., 2014). This produces a more spatially complete SDS over both land and ocean. The merged MODIS AOD product uses DB data over surfaces where the Normalized Difference Vegetation Index (NDVI) $\leq 0.2$ and DT data where NDVI $\geq 0.3$. For intermediate NDVI regions, the algorithm with the higher quality assurance flag is used. Sayer et al. (2014) provide a detailed analysis of these products and note that DB performance is poorer over dusty regions, compared to the global average, with an overall tendency to underestimate AOD in dusty environments. Additionally they find that in the Sahel, contributions to AOD from different aerosol types are likely to contribute to frequent different AODs retrieved by the two algorithms, though DB performs better than DT in this region. Therefore when the merged SDS draws data from the DT SDS, the quality is reduced in the Sahel. Here we present the merged MODIS SDS since it provides a more continuous dataset for comparison over the SWAMMA domain than simply the DB SDS. We show data from the Terra satellite with a 10:30 LST overpass, L3 monthly mean data with a spatial resolution of 1 degree. We note that there are anomalously high MODIS merged AODs present in Fig 2 in June around 0-10N, 15-30E (bottom right corner – southern Sudan and Central African Republic), and to some extent in this region in July as well. These high AODs are not present in the DB SDS (not shown) as they originate from the DT SDS (not shown). These anomalies have been identified as a result of an AOD dependence on solar angle investigated in detail in Wu et al., (2016). We therefore consider this region of high AOD to be an artefact of the DT contribution to the merged SDS, which in this particular case is likely to be less reliable due to the reasons explained above.

### 2.2.2 SEVIRI RGB dust imagery

False colour Red-Green-Blue (RGB) dust imagery from the EUMETSAT Spinning Enhanced Visual and Infrared Imager (SEVIRI) is used to give a qualitative understanding of the uplift of dust associated with a large cold pool. The 15-minute time resolution and very wide field of view makes SEVIRI data extremely useful for visual tracking and interpreting the development of individual systems. To highlight regions of raised dust the product compares brightness temperature and



brightness temperature differences between three of SEVIRI's infrared channels (channels 7, 9 and 10 which correspond to 8.7, 10.8 and 12 µm wavelengths respectively). While the magenta colour associated with raised dust can be indicative of important dust uplift mechanisms there are several limitations to its use. These include biases caused by the height of the dust layer, the lower tropospheric lapse rate and masking of lifted dust by high column water vapour (Brindley et al., 2012).

### 2.2.3 SEVIRI AERUS-GEO AOD

The AERUS-GEO (Aerosol and surface albEdo Retrieval Using a directional Splitting method- application to GEOstationary data) AOD is a daily daytime only mean measure of AOD (Carrer et al., 2014). The approach to produce the AERUS-GEO product is detailed in Carrer et al., (2010) and Carrer et al., (2014). The relatively invariant nature of the land surface albedo on a daily timescale compared to the atmosphere is used along with the high temporal resolution of SEVIRI retrievals (full disc scan every 15 minutes) to distinguish the 0.63 µm signal from aerosols from that of the surface. The AERUS-GEO product has good accuracy when compared with other satellite derived AOD products (typically less than 20% deviation from AERONET) and has much better spatial and temporal coverage than products that utilise data from polar orbiting satellites.

### 2.3 Surface wind observations

Wind speed observations from several in situ observation platforms are used to compare with simulated wind speeds. Data from 5 stations are used, these are: Fennec automatic weather stations (AWSs) 134 (23.5 °N,3.0 °W) and 138 (27.4 °N, 3.0 °W), the Fennec flux tower deployed at Bordj Badji Mokhtar (BBM; 28.3 °N,0.9 °E) and African Monsoon Multidisciplinary Analysis (AMMA) "Couplage de l'Atmosphère Tropicale et du Cycle Hydrologique" (CATCH) AWSs at Agoufou (15.3 °N,1.5 °W) and Kobou (14.7 °N, 1.5 °W). Different dust uplift mechanisms occur at different times of day, the advantage of these observations compared to routine synoptic observations is their high temporal resolution (which allows for resolution of the diurnal cycle) as well as the geographical spread of stations across the Sahel and Sahara, which are generally very poorly observed. For comparison between simulations and observations taken at different heights all winds are adjusted to 2 m height using the wind profile power law $u = u_r (z/z_r)^\alpha$ where $u_r$ is wind speed reference height ($z_r$), z is the height to be adjusted to, and α is a stability coefficient (nominally 0.143; Touma, 1977, Roberts et al., 2017).

### 2.3.1 Fennec AWS

The Fennec project aimed to improve the understanding of Saharan meteorology with a particular focus on the processes associated with dust uplift and transport. Eight Fennec AWSs were distributed across the Sahara in Algeria and Mauritania in late May 2011 and continued to operate into 2013. The structure of the AWSs and the observations that were made are detailed in Hobby et al., (2013). Unfortunately during 2011 a number of the AWSs experienced problems associated with overheating





leaving only F-134 and F-138 with good data coverage over the SWAMMA simulation period (Roberts et al., 2017). Wind observations were transmitted via satellite and comprised 3 minute 20 second mean wind speed values from the cup anemometers at 2m above ground level.

### 2.3.2 Fennec BBM supersite

Also deployed as part of the Fennec campaign was a more comprehensive suite of instruments at 2 supersites at BBM (Algeria) and Zourate (Mauritania). The wind speed observations that are used in this study are from the flux tower deployed at BBM (Zourate data does not extend sufficiently throughout the simulated period). The supersite has no wind speed data for May but has data for 25 days in June, 31 days in July, 31 days in August and 3 days in September. This allows for comparison between simulations and observations for 3 of the 5 simulated months within the West African summertime dust hotspot (Englestaedter and Washington 2007, Knippertz and Todd 2010). Marsham et al., (2013) details the instrumentation deployed at the BBM supersite. Winds measurements used in this study are from a sonic anemometer positioned 10 m above ground level. The sampling frequency is 20 Hz but 1 hour means have been calculated for comparison with simulations and other observed winds.

### 2.3.3 AMMA-CATCH stations

The AMMA field campaign (Lebel et al., 2011), primarily conducted in 2006, had the aim of improving the understanding of the WAM system. Observations over a large area and over a large timescale were conducted including the deployment of AWSs. Of the many AWSs deployed 2 of those have been used in this study and were part of the AMMA CATCH program, which specifically had the objective of looking at interannual variability of the WAM system. These stations (Agoufou and Kobou), were deployed ready for the main AMMA observing period in 2006 and were still operational in 2011. This allows for unprecedented comparison between simulations and observations in the Sahel and Sahara, with observations that are temporally coincident.

### 2.4 Storm tracking

To investigate the nature of mesoscale convective systems seen in observations and those generated in convection permitting simulations a storm tracking approach has been adopted. The algorithm used is based on that of Stein et al., (2014) and has been modified for use on both simulations and observations (Crook et al., in preparation). The algorithm can be applied to either rainfall or brightness temperatures to track convective systems over West Africa. Storm clusters are identified through the use of a threshold and by grouping contiguous cells. These are then followed in time using a fractional overlap method (0.6 overlap threshold) to track storm cells allowing for both cell splitting and merging. If a storm has no overlapping cells from




the previous time-step, then it is a new initiation. When a storm has no overlapping cells in the next time-step it is a dissipation. For splits the cell with the greatest overlap retains its storm ID while other cells are said to have split and are given new storm IDs (parent IDs are recorded). Similarly, for merging, the cell from the previous time-step with the greatest overlap with the resultant cluster maintains its ID and any other cells with smaller overlaps are said to have merged and take the ID of the cell

with the largest overlap. For this study it was decided that a brightness temperature approach, using a threshold of -40 °C would be best suited. This is due to the use of hourly data for tracking, where clouds give a greater overlap and therefore chance of tracking between time-steps. This is also because we are interested in systems where rain both does and does not reach the surface, since both can produce haboobs, making rainfall tracking less reliable. Therefore for this study hourly brightness temperatures calculated from both simulated and observed (SEVIRI channel 9, 10.8 μm) outgoing longwave

radiation have been used to track systems, giving information about storm triggering, locations, size and storm lifetime.

## 3 Results

### 3.1 Impact of resolving convection on dust AOD and dust emission

Comparison of dust AODs with observations is frequently used to verify (and in many cases, tune) dust models (Huneeus et al., 2011, Huneeus et al., 2016). This is because AOD observations from satellite are now available at high temporal and spatial

resolution, unlike observations of dust emissions and concentration. However, within a modelling framework AOD is very much an end product, requiring not only accurate representation of all the physical processes involved in dust emission, transport and deposition to achieve realistic dust loadings, but also accurate representation of particle size distribution and spectral optical properties. For example, in the SWAMMA experiments, although extinction per unit mass is greatest for particle size division 2 (0.1-0.3 μm mean radius), dust mass is maximum in division 4 (1-3 μm mean radius), and total

extinction for dust is dominated by particles in size division 3 (0.3-1 μm mean radius). On the other hand models with very similar AODs can have very different dust emissions, due to compensating differences in deposition, transport or particle size distribution (Kinne et al., 2003; Ocko and Ginoux 2016; Evan et al., 2014).

The dust loadings in the SWAMMA experiments (5-6Tg May to September seasonal mean for the whole domain) are at the low end of, but not outside, the range reported by other modelling studies (this of course could be resolved by tuning total emissions, but would not affect the systematic model biases we investigate here); Huneeus et al., (2011) reviewed fifteen

global models within the AeroCom project and found global loadings ranged between 7-30 Tg , of which ~70% have been estimated to be attributable to the Sahara (Luo et al., 2003). All versions of the model here are initialised with zero dust and found to be spun-up within 5-10 days; for ease of analysis and consistency with presentation, monthly means for May are presented here for the whole month with no special treatment of the spin-up period (it should be noted that even including spin

up May has dust and AOD values in excess of any other simulated month).



Figure 2 displays the monthly mean (May-September) AODs at 550nm from the MODIS Terra satellite with the dust AOD from all the SWAMMA models excluding dust radiative effects (4E, 12E, 12P and 40P from Table 1). Here the model AOD at 1000 UTC has been selected to provide a better time match for the Terra data which overpasses the region at approximately 1030 UTC. It is clear that the models are very similar across all resolutions and all feature a maximum over the

Bodélé depression (~ 18°N 19°E) in all months, in common with the MODIS data. However, apart from in May, the models all have insufficient dust over the central Sahara and a strong maximum over the west coast at ~ 20°N which is not evident in the MODIS data. These are common features of many models (e.g. Figure 2 of Todd and Cavazos-Guerra, 2016, Figure 6 of Ridley et al., 2012) and, as pointed out by Evan et al. (2014) in a multi-model CMIP5 comparison study, may have many contributory factors including poor representation of soil texture, moisture and vegetation cover, and deficiencies in model

surface winds. The focus of this study is to see if any improvement can be achieved by resolving convection explicitly, since haboobs are known to be a key uplift mechanism in the summer-time central Sahara. Figure 3 therefore compares the spatial correlations and biases (at model grid-points) for the 12km simulations with explicit and parameterised convection (12E and 12P) relative to the MODIS AODs, broken down into specific regions as shown by the boxes in Figure 1b;, North Sahara (NS, 25°N-30°N), the Sahara (SA,15°N-25°N), the Sahel (SL, 10°N-15°N) and the Guinea Coast (GC, 5°N-10°N). Overall, where

they are significant (> 0.5) correlations are positive, except for the Guinea Coast region in June where dust loads are lower and as noted in section 2.2.1 MODIS data are anomalous. Correlations are high (> 0.5) for the North Sahara throughout the season, and also in May in the Sahel, and lower at the other locations/times, which are when moist convection and haboobs are known to be most active. Differences between the explicit and parameterised versions of the model are small, with the parameterised version generally having slightly better correlations with MODIS except for July-September in the Sahara. Despite the low

correlations in the summertime Sahara, this is the region where we would look to find improvements in dust in the convection permitting simulations, this is due to the expectation that in this region there are regions with surface characteristics that allow for the deflation of dust, as well as the additional uplift process expected to be represented (haboobs). The model AOD biases relative to MODIS data are predominantly negative and have the greatest magnitude in the south of the SWAMMA region, consistent with the model producing too little dust there (although the maximum bias of ~ -0.55 in June in the Guinea Coast

region is where the MODIS data are anomalous). Exceptions to this are the North Sahara and Sahara in May, where biases are positive, but small, suggesting too little dust in regions/seasons when moist convection is most active but too much prior to the monsoon onset and close to the Atlantic coast. Where there are differences between explicit and parametrised simulations, the explicit version mostly has larger biases than the parameterised model, although the differences are small.

All the SWAMMA models are lacking the AOD maximum evident in the MODIS data from June to August in the

central Sahara. We therefore examine factors affecting the dust emission to see why this might be. Figure 4 shows the monthly mean (May-Sept) dust AODs (for all hours), with the corresponding dust emissions, surface friction velocity over bare soil (U*) and soil moisture in the top 10cm soil layer for the 12km explicit convection model (12E). Reference to the clay fractions in Figure 1 is also helpful as it is a factor in the vertical dust flux equation. Areas with high clay fraction (up to a maximum value of 0.2) have the potential to produce the most dust in dry conditions, however it is also the case that high clay soils are



more sensitive to soil moisture, with higher soil moisture values impeding emission. South of approximately 15 °N emission of dust is negligible due to the very small bare soil fractions (see Figure 1) and higher soil moisture values (Figure 4) ; although the JULES surface exchange scheme includes a seasonal climatology of fractional leaf area index (LAI), the fraction of each land type, including bare soil, is fixed. It is known that there is a strong seasonal cycle in vegetation over the Sahel (Mougin

et al., 2009) with summertime dust emission from haboobs during the early monsoon season (June-August; Klose et al., 2010, Knippertz & Todd 2012), and even cold pools from congestus clouds can lead to visible dust uplift here in June (Marsham et al., 2009). This fixed bare soil fraction therefore means that seasonal dust emission from the Sahel cannot be realistically represented in this configuration of the UM. Figure 4 also shows that in the model emissions are strong over the Bodele, the west coast and central Algeria where the highest U* values coincide with regions of high clay fraction and low soil moisture.

To see the impact of the choice of convection scheme, Figure 5 shows the differences between the 12km explicit and parameterised models for the same variables. There is a clear mid-season switch in the AOD, emissions and U* 12E-12P differences such that they are generally (over the whole SWAMMA area) much more negative July-September than May–June. Differences in soil moisture have relatively little impact on the dust emissions because they are mainly to the south of the region, where the bare soil fraction is small. However, it should be noted that the soil moisture available to inform the

emission scheme is the top 10 cm mean soil moisture; this is likely to have a buffering effect on emission as in reality the skin soil moisture controls dust emission. This has a much faster timescale for drying and reaching a level appropriate for mineral dust deflation: Gillette et al., (2001) reports uplift of sediment from a dry lake in California being raised 10-30 minutes after rainfall and Bergametti et al., (2016) reports of Sahelian surfaces taking less than 12 hours to fully recover their dry sand transport potential. Some areas of increased U* and dust emissions are evident for the 12E model in the central Sahara (~

20°N, 0°E) in June, and to a lesser extent in July, but these do not produce any overall decrease in the model-MODIS bias for the SA region in Figure 3 due to compensating increases elsewhere.

Figure 6 summarises the seasonal trends in factors affecting the dust AOD in the 12E and 12P models for the North Sahara (NS), Sahara (SA) and Sahel (SL) regions. We see that for both simulations the AODs are all at their highest in May, whereas for MODIS it increases from May to a maximum in July (for NS and SA) and June (for SL), consistent with the

summertime northwards advance of the monsoon, rainfall and haboobs (Marsham et al., 2008). MODIS AOD data from 2006-2008 in Figure 2 of Ridley et al. 2012 indicate that this pattern is robust and not unique to 2011, indicating that simulations are missing a key dust generating mechanism providing a maximum in June-July. Additionally we see that the explicit convection version generally performs worse than the parameterised in this respect. Analysing the contributory factors, the trend in model AOD follows the trend in dust load, as expected (note that loads plotted are regional totals scaled by a factor of

5 for NS and SA, and 10 for SL). The dust loads generally follow the trend in dust emissions, except for May-June in the Sahel where the dust load is boosted by advection from the Sahara. Dust emission trends are strongly driven by the friction velocity (U*) in NS and SA throughout the season, where soil moisture values are too low to have much influence (except for SA in August where the monsoon rains encroach on the region). For the Sahel the pattern is different however, with lower U* and much higher soil moisture values combining to drastically reduce emissions and dust loads as the monsoon season evolves. In





the Sahara and Sahel friction velocity values are generally lower, and soil moisture values higher in the convection-permitting than the parameterised model, leading to lower dust emissions, loadings and AODs (Since the monsoon is further north in the explicit run, not shown but consistent with Marsham et al., 2013 and Birch et al., 2014). For the North Sahara the explicit version has lower soil moisture and higher U* than the parameterised in May-June leading to higher dust AODs which exceed

the MODIS values; however, this is not sustained through the rest of the season and AOD biases are negative for July-September.

The explicit treatment of convection is known (from Cascade, Marsham et al., 2011 and Heinold et al., 2013) to have a strong impact on the representation of haboobs, but here does not impact the dust fields significantly. We therefore continue our investigation with an evaluation of the model winds against observations, to further explain why explicitly permitting

haboobs has such a small impact on the modelled dust AODs.

## 3.2 Impact of resolving convection on dust-generating winds

The hypothesis that explicit convection would produce significant differences in the dust field has been shown to be incorrect. One potential cause of this is the possibility that the simulated surface winds do not change very much from one simulation to another. Figure 7 shows the distribution of wind speeds at a number of locations in the Sahel and Sahara for all four simulations

(4E, 12E, 12P and 40P) as well as observed winds. There is close agreement in the maximum frequency of occurrence in the simulations at each of the station, with the observations up to 3 m/s lower, with the largest differences in the Sahel. The advantage of showing the distributions on a logarithmic y-axis is that the frequency of rare high wind speed events can be examined, while the frequency of such events might be low, the non-linear nature of wind speed to dust uplift (above a threshold) mean that they dominate dust uplift (Cowie et al., 2015). This highlights that F-134 and BBM in the southern Sahara

both have observed high wind speed events that are significantly under-represented in any of the simulations. This is particularly important as both these stations are located in a region where haboobs are known to be significant and in the seasonal maxima of AOD that can be seen in Figure 2 (top row), but is absent in all SWAMMA simulations (and CMIP5 simulations).

In order to investigate haboob winds, Figures 8a and 8b show the anomaly of 10-m wind speed cubed composited

around column maximum rainfall rates greater than 1 mmhr$^{-1}$ for the 12 km simulations with parameterised and explicit convection (over the region 15 °W to 20 °E, 10 °N to 30 °N, where it is expected that if a cold pool were to occur it could feasibly raise dust). The anomaly is calculated as the difference from average wind speed cubed values calculated for each simulated month and time of day, to reduce the effects of the seasonal and diurnal cycles. The period for the composite average covers the time at which the rain threshold is met and the following six hours. This highlights the production of convective

cold pools in the convection-permitting simulation which have wind speed cubed values in excess of the average for that time of day and season. The peak in the centre of the composite domain in Figure 8b and the absence of a peak in Figure 8a indicates that cold pools are indeed present in the 12 km explicit convection simulation and that these features are absent in the



parameterised version. Another important feature of the cold pool anomaly shown in Figure 8b is the extent of the positive anomaly field. Although this cannot provide a direct measurement of the size of cold pools in the simulation it clearly indicates that they can reach very large sizes (in excess of 300 km radii). With cold pools of this size it might be expected that there would be a noticeable impact on the uplift of dust and therefore the distribution of AOD. Figures 8c and 8d are the same type of composite as Figs. 8a and 8b but for the maximum 10 m wind speed recorded within the rainfall plus 6 hour window described above. This gives greater information about the actual strength of the winds generated by the presence of convectively generated cold pools. The maximum wind speed composite for the parameterised convection simulation (figure 8c) indicates generally weaker winds than that seen in the explicit simulation. However, the maximum cold pool winds (which clearly show a positive anomaly in figure 8b) are relatively weak, reaching a maximum composite values of between 6 and 8 ms$^{-1}$. This is lower than the mean maximum wind seen in Provod et al. (2016) for observed Sahelian cold pools of 8 to 10 m/s. As the values in the centre (and just north of centre) of the domain would be most likely to be affected by almost all cold pools generated it would be expected that there would be only minor effect of reducing the wind speed values via compositing. With this in mind, it is surprising that the maximum value measured would be approximately 8 ms$^{-1}$ as this represents only a minor exceedance of (or even a failure to exceed) the approximate 7-8 ms$^{-1}$ dust uplift threshold used in many emission schemes (Marticorena et al., 1997). This suggests that although there are clearly cold pools being generated in the convection permitting simulation, and these cold pools produce anomalously strong winds, they are not as strong as might be expected and certainly the strongest cold pools winds, which are known to generate the extreme winds in the southern Sahara and Sahel stations are missing in the models (Figure 7).

The unchanging overall frequency of different wind speeds (Figure 7) and the presence of convectively generated cold pools (Figure 8) combined with the findings of Marsham et al. (2013) that up to 50% of dust emission in the summertime central Saharan hotspot occurs at night due to haboobs highlights the need to compare diurnal cycles of the different simulations. Figure 9 shows the diurnal cycle of dust uplift potential (DUP, Marsham et al., 2011) for all simulated months for the 5 sites for which there are observations. The Northern Sahara station, F-138, has a similar development of the diurnal cycle across the 5 simulated months: in both observations and simulations the highest DUP values tend to occur during the day with much lower values at night, and in some months there is at 0900 UTC maximum, likely from the breakdown of the nocturnal LLJ. This is as expected given that F-138 is too far north to be strongly or regularly influenced by the cold pools spreading deep into the Sahara. The low night time values reflect the development of a stable nocturnal boundary layer, which breaks down due to surface heating during daylight hours. F-134 and BBM in the Saharan box show a clearer peak from LLJ breakdown at approximately 0900 UTC. At F-134 (in both observations and simulations) this process is the dominant feature throughout the entire season. However, further south at BBM the observations suggest that the morning peak in DUP is similar in magnitude with an evening peak, in agreement with Marsham et al., (2013). This second peak in DUP associated with haboobs is not well represented in the simulations with the 12 km explicit and 4 km explicit simulations having different diurnal cycles with regard to the evening peak. This is possibly caused by the simulations failing to produce cold pools of sufficient strength as far north as BBM. However, the evening peak at BBM cannot be wholly attributed to cold pools. This is



due to the fact that there is a similar, yet smaller, peak present in the simulations with parameterised convection in June. This is feasibly the impact of the daily night-time surge of the monsoon flow which is stronger in the parameterised simulations then the explicit (consistent with Birch et al., 2014).

In the Sahelian stations of Agoufou and Kobou the diurnal cycle in May (Figs. 8p and 8u) is similar to that seen in the Sahara with a morning LLJ peak in DUP and largely similar diurnal behaviour across all simulations. However, by June there is evidence of divergent behaviour between the simulations. At Agoufou and Kobou (Figs. 8q, 8v) there is an evening peak in DUP at 1600-2100 UTC which is more pronounced in convection permitting simulations. This evening peak grows more pronounced at these stations through July into August. This evening peak is also particularly noisy: this behaviour is what would be expected from high DUP values associated with cold pools due to their production of very high wind values that last on timescales <1 hour (for observations at a fixed point). When combined these features mean that an average diurnal cycle produced over a relatively short period of time (1 month) will not produce a smooth evening peak in DUP. Concomitantly the convection permitting simulations also have a reduction in size of the morning NLLJ DUP peak (consistent with Marsham et al., 2011). As shown earlier in Figure 7 there is little change in the overall distribution of modelled wind-speeds with explicit convection, showing how overall the increased evening winds are compensated for by the decreasing morning winds.

Given that it has been shown that convective cold pools are present and are likely to be responsible for a significant modification of the diurnal cycle of winds in the Sahel and as far north as BBM it follows that there should be some modification in the uplift and transport of dust. Figure 10 shows the monthly mean diurnal cycle in dust emissions from the 12km simulations for the 5 stations. Although there is some evidence of an evening increase in emissions in the convection-permitting model at BBM in June-August, consistent with the DUP in Figure 9, this is insufficient to significantly change or improve the dust load and AOD. Dust emissions at the stations in the Sahel (Agoufou and Kobou) are reduced in the explicit version; this is likely to be due to the increased soil moisture in that region (as demonstrated in Figure 6). In addition to such limits imposed by the surface characteristics on the uplift of dust in the model, it is also possible that there is some behaviour of convective storms and their associated cold pools that means that they do not lift dust; for example the wrong size, lifetime, or location. This is examined in the next section.

## 3.3 Impact of resolving convection on modelled storms

To investigate the nature of the storms that are responsible for the generation of cold pools a storm tracking approach has been used. This takes advantage of the availability of satellite observations of outgoing longwave radiation from which brightness temperature can be easily derived, and tracking is performed on features with a brightness temperature below -40 °C. This means that direct comparisons of the storms produced in the convection permitting simulations can be made with those identified through satellite retrievals. To reduce the number of events that were considered and to highlight the impact of larger events which dominate observed dust uplift in the central summertime Sahara (Marsham et al., 2013; Allen et al., 2013), only storms that reached a threshold value in size (approximately 5,000 km$^2$) were considered. These systems will be referred to as





mesoscale convective systems (MCSs) hereafter. The total number of MCSs between the 4 km and 12 km convection permitting simulations and the observations is not dissimilar having 14,082, 15,555 and 12,843 respectively; however, the storm track densities (number of times a storm track is centred over a specific region on a 0.25 degree grid) shown in Figure 11(a)-(c) shows that there is a greater density of events in both of the simulations compared to the observations. This is likely due to the generally enhanced lifetime of MCSs seen in simulations (Figure 11e). The spatial distribution of MCS track density (based on storm centres) indicates that their latitudinal position in simulations compared to reality is approximately correct, with MCSs (and therefore cold pools) occurring commonly as far as 17 °N, indicating that the positioning of the MCSs is not the driving factor behind the lack of dust raised here. However, the higher frequency of very large storms in SEVIRI imagery compared to explicit simulations (Figure 11(d)), the generally weak cold pool winds identified in Figure 8d and the missing tail of high winds in Figures 7b and 7c, suggest that the region affected by large cold pools has an under-representation of cold pool winds in convection permitting simulations.

Figure 11e shows the distributions of MCS duration (to the nearest hour), highlighting the fact that the 4 and 12 km simulations have MCSs that last longer on average than those in SEVIRI; it is only storms that live beyond 30 hours for the 4 km simulations and 47 hours for the 12 km simulations that the frequency of occurrence first drops below the values seen from observations. This abundance of events (even MCSs that are smaller than those observed) and the fact that convective cold pools are clearly being produced in the simulations (despite their reduced strength) suggests that the lack of emission in simulation south of 17 °N cannot, however, be entirely attributed to smaller MCSs producing smaller/weaker cold pools.

In interpreting the storm-track based analysis discussed above, it is informative to examine sample images of observed and modelled large storms. Figure 12 is a case study of a large cold pool event that occurred on 23rd August 2011. It was well represented in the 4 km simulation in that the timing and location of initiation of the system was roughly correct, after which a large MCS developed and produced a cold pool which spread north and west into the Sahara. Although we do not necessarily expect an accurate one-to-one correspondence between observed and modelled storms, this far into the simulation, the case shown does share key similarities and is one of the larger modelled storms from the simulated period. The cold pool in the simulation can be seen through both the elevated friction velocity over bare soil, as well as the spreading of air away from the MCS shown in the 10 m wind vectors. Similarly, the cold pool generated in reality can be identified through the occurrence of arc clouds along the leading edge of the cold pool and the magenta colour that identifies raised dust within the cold pool in the SEVIRI RGB false colour dust images. The impact that this cold pool has on dust is assessed through the daytime averages of the dust AOD from the 4 km simulation and the SEVIRI AERUS-GEO AOD product. There is clearly a strong AOD signal associated with the cold pools in both measures. However, the signal in the simulation is dwarfed by the high levels over the western part of the domain (at least partially associated with erroneously high uplift over the western Sahara region). Whereas, in the SEVIRI AERUS-GEO product the AOD feature in the central Sahara is comparable in magnitude to the transported plume over the Atlantic and is much more clearly linked to uplift caused by strong near-surface winds associated with the



passage of a convective cold pool. This is consistent with the maximum mean hourly observed wind on this day at BBM being 11.4 ms$^{-1}$, and the maximum instantaneous modelled cold pool wind being 7.5 ms$^{-1}$.

## 4 Conclusions

We have investigated whether biases in dust AOD over the Sahara and Sahel, known to exist in many global and regional
models, can be improved by using an explicit rather than parameterised formulation of convection. It was hypothesized that explicit resolution of the strong winds associated with cold-pool outflows which generate dust storms (haboobs) in summertime West Africa might enhance the AOD in the central Saharan Heat Low (SHL) region, where haboobs have been observed to be a key uplift mechanism, and where a dust maximum is present in satellite retrievals but missing in many models. Regional versions of the Met Office Unified Model with prognostic dust at 4 km, 12 km and 40 km grid-spacings were used, with
explicit convection at 4 km and 12 km, and parameterised convection at 12 km and 40 km. These SWAMMA simulations enable a clean comparison between models at 12 km resolution with explicit and parameterised convection (differing only in representation of convection). This provides a "seamless" approach, with the model configurations ranging from high resolution (4 km) convection-permitting to a configuration similar to a climate model. In this respect a potentially valuable property of the SWAMMA simulations is their similarity in behaviour and AOD features with CMIP5 simulations, indicating
that investigation of process errors in SWAMMA are likely to identify and provide knowledge about similar errors in the CMIP5 dataset.

The results show that all SWAMMA simulations have very similar dust AOD fields, despite explicit convection significantly changing the wind fields and overall clearly demonstrate how improving the representation of cold pools, known to be critical to dust uplift, is a necessary but not sufficient condition for improving AOD fields. When convection is modelled
explicitly cold pools (haboobs) are present and the diurnal cycle in surface winds is better represented. However, in the southern Sahara the rare very strong wind events that result from haboobs and cause the most intense dust storms are still absent in all simulations. Analysis of composite cold pools and storm tracking shows that although storms exist far enough north in the in convection permitting simulations, the storms are not sufficiently large, which is likely to limit both the intensity of the cold pool winds and the northwards propagation of the resultant cold pools into the southern Sahara, and so is consistent with the
weaker than observed winds in that key region. This interpretation is supported by a simple representative case-study of a large storm that shows how in the model even when a large system is generated it does not raise quantities of dust comparable to those seen in satellite retrievals. Consistent with past studies of long-duration large-domain runs, in the explicit runs there is a reduction in the strength of the morning low-level jet (LLJ), which compensates for the haboob uplift. This means that the increase in dust emissions achieved by the strengthened evening (haboob) winds does not produce any overall increase in the
AOD in the SHL region, since the LLJ winds are reduced. The results here likely contrast with those of Chaboureau et al. (2016), where explicit haboobs did improve dust fields for several reasons: i) in the Chaboureau setup it is not expected that the explicit convection weakens the low-level jet because their simulations are initialised daily and run for between 24 and 72



hours depending on model (as seen in comparisons between 2-day and 10-day runs in Marsham et al., 2011), ii) the Chaboureau models have a different land surface to the UM and different dust emissions schemes which are individually tuned so that AOD changes cannot be attributed solely to the choice of convection scheme and iii) the Chaboureau models with explicit convection have a more limited southern boundary than SWAMMA simulation so their results are more focussed on Saharan,

rather than Sahelian dust (results are shown for the region 13 °N-31 °N).

The results here also suggest several key problems with the modelled land surface. The models have almost no dust uplift in the Sahel, whereas in reality convective storms over the Sahel do raise dust (Flamant et al., 2007; Marsham et al., 2009; Roberts and Knippertz 2014). South of 15 °N the models have a low and temporally unvarying bare soil fraction which is unable to release sufficient dust even if surface conditions and winds are favourable; in reality it is known that the Sahel has

a large variation in bare-soil fraction seasonally and interannually (Mougin et al., 2009). The use of soil moisture in the model is also implicated, since the model uses soil moisture over a 10cm layer, whereas in reality it is the skin soil moisture that is relevant and both the soil makeup (sandy soils) and the hot, dry conditions in the northern Sahel and Sahara mean that the actual time between rainfall and dust emission can be much shorter than that predicted by the SWAMMA simulations (Gillette et al., 2001, Bergametti et al., 2016). This role of the land-surface errors in the Sahel is consistent with recent analysis of

operational global UM runs (Pope et al., 2016). Finally, clay fraction is a crucial soil texture parameter in several of the dust emission and flux calculations and high clay fractions over the west coast in combination with strong northerly winds blowing off the Atlantic cause high AOD values there which are not seen in observations.

The issues discussed above provide a stark demonstration of the number of marginal processes that must be well simulated in any model to capture the seasonal evolution of the dust field over Africa. Models must capture: the seasonal

evolution of the continental-scale thermodynamics gradients, itself non-trivial and dependent on convection (Marsham et al., 2013); the location of the moist convection, particularly the marginal convection close to both the leading edge of the monsoon and close to the sharp gradient in soil moisture and vegetation present from the Sahel to the Sahara; the tail of strong winds from cold pools and the low-level jet breakdown; the time-evolution of skin soil-moisture and vegetation (and therefore roughness); and the soil properties themselves. Given these challenges it is perhaps not surprising that Evan et al (2014)

conclude that the CMIP models are unable to capture any of the salient features of north African dust emission and transport. An improved representation of cold pools in dust models is clearly necessary, but not in itself sufficient for improving AOD fields. Future evaluations of dust models should ensure that winds as well as dust are evaluated to ensure that models are getting the right answer for the right reasons (noting the value of observed not analysed winds, due to the large biases in analyses). Although parameterisations of haboobs (e.g. Pantillon, 2015; 2016) are clearly valuable, corresponding

improvements are also needed in soil moisture, vegetation and soil properties in models. It is also clear that winds from explicit models may still have significant biases even though haboobs are represented, and therefore estimates of the fraction of dust uplift from haboobs from such models (e.g. Heinold et al., 2013), although very valuable, may be a significant under-estimate and must be treated with caution.



**Acknowledgements**

The SWAMMA project was funded by the UK Natural Environmental Research Council (NERC) standard grant
NE/L005352/1. John Marsham was also funded by AMMA 2050 (NE/M020126/1), IMPALA (NE/M017176/1) and
DACCIWA (FP7/2007-2013 under grant Agreement no.603502). This work used the ARCHER UK National Supercomputing
Service (http://www.archer.ac.uk) to perform the model experiments and    the JASMIN super-data-cluster
(doi:10.1109/BigData.2013.6691556) at the Centre for Environmental Data Archival (CEDA) for longer-term storage and
analysis of model output.  The assistance of G. M. S. Lister at NCAS-CMS Reading in facilitating the model runs is gratefully
acknowledged. We also thank S. Woodward (UK Met Office Hadley Centre) and C.E. Birch (University of Leeds) for their
advice with various aspects of the model setup. MODIS AOD analyses used in this paper were produced with the Giovanni
online data system, developed and maintained by the NASA GES DISC. We are grateful to EUMETSAT for SEVIRI data
used for storm tracking. SEVIRI RGB dust imagery is available from http://www.fennec.imperial.ac.uk and SEVIRI AERUS
GEO AOD imagery is available from the ICARE Data and Services Center www.icare.univ-lille1.fr. The Fennec AWS
network was developed, tested and installed as part of Fennec project (NE/G017166/1). AMMA-CATCH system was funded
by the French Ministry of Research and National Institute for Earth Sciences and Astronomy.

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





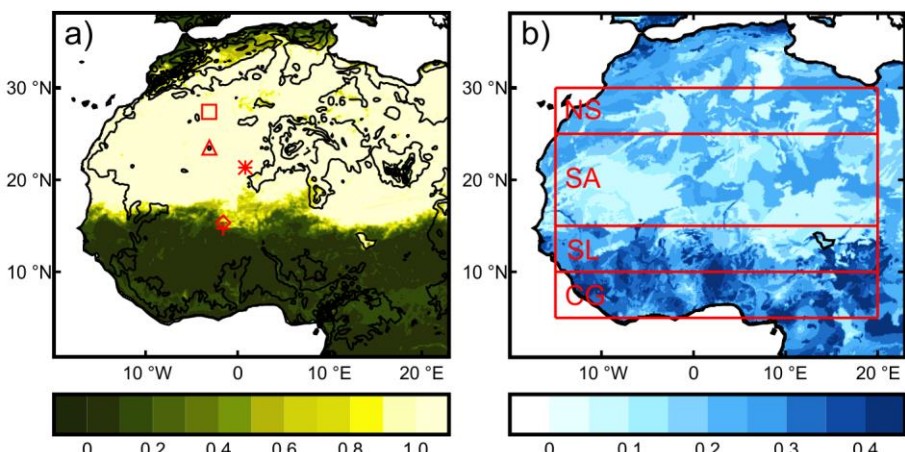

**Figure 1.** Maps of a) bare soil fraction for 12km models (filled contours) with orographic height (open contours from 0.2 km with 0.4 km interval). Locations of Fennec and AMMA stations marked in red: square F138 (27.4°N 3.0°W), triangle F134 (23.5°N 3.0°W), asterisk Bordj-Badji-Mokhtar (21.3°N 0.9°E), diamond Agoufou (15.3 °N 1.5°W) and cross Kobou (14.7°N 1.5°W) and b) clay fraction from Harmonised World Soil Database (HWSD) used to define surface soil texture in the models. Red boxes highlight regions referenced in Fig 3 [North Sahara (NS, 25°N-30°N), the Sahara (SA,15°N-25°N),  the Sahel (SL, 10°N-15°N) and the Guinea Coast (GC, 5°N-10°N), all with longitudes 15°W-20°E].





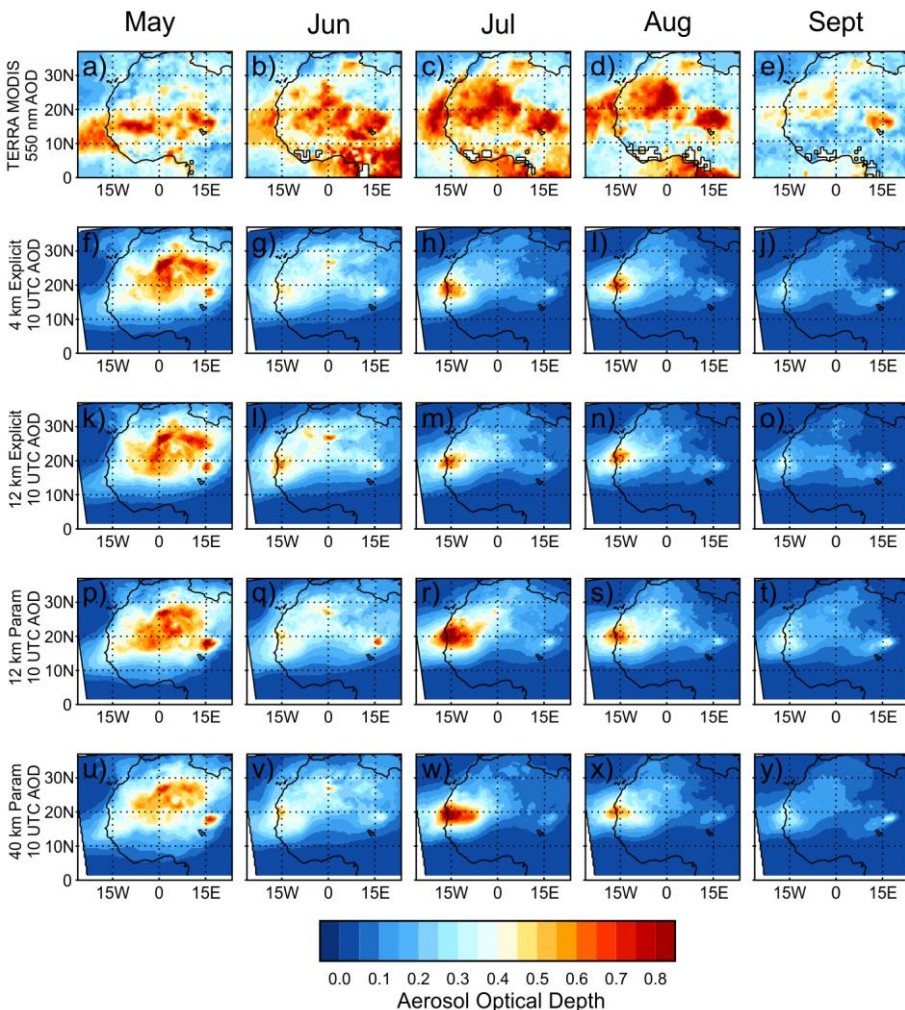

**Figure 2. Monthly mean (May-Sep, left-right) Aerosol Optical Depths (AOD) at 10 UTC from: a) to e), MODIS Terra satellite (combined Deep-Blue and Land-Ocean datasets), f) to j) 4km simulation with explicit convection (4E), k) to o) 12 km simulation with explicit convection (12E), p) to t) 12 km simulation with parameterised convection (12P) and u) to y) 40 km simulation with parameterised convection (40P).**





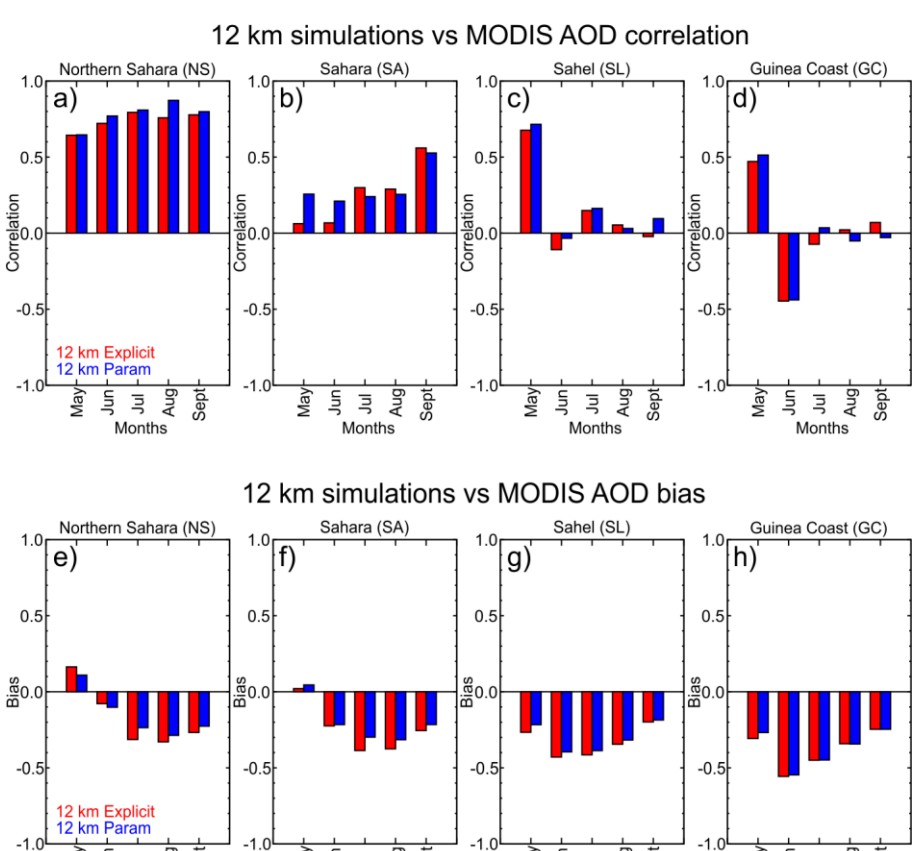

**Figure 3. Aerosol Optical Depth (AOD) correlation coefficients and biases between 12 km simulations (explicit and parameterised 1000 UTC) and MODIS AOD retrievals (~1000 UTC). a) to d) monthly mean (May-Sept) model AOD vs MODIS AOD correlation coefficients. e) to h) monthly mean (May-Sept) model – MODIS AOD biases. Shown are 12 km explicit convection simulation (12E, red) and 12 km parameterised convection simulation (12P, blue). Correlations and biases calculated from boxes shown and labelled in figure 1 [a) and e) Northern Sahara (NS) box, b) and f) Sahara (SA) box, c) and g) Sahel (SL) box and d) and h) Guinea Coast (GC) box].**




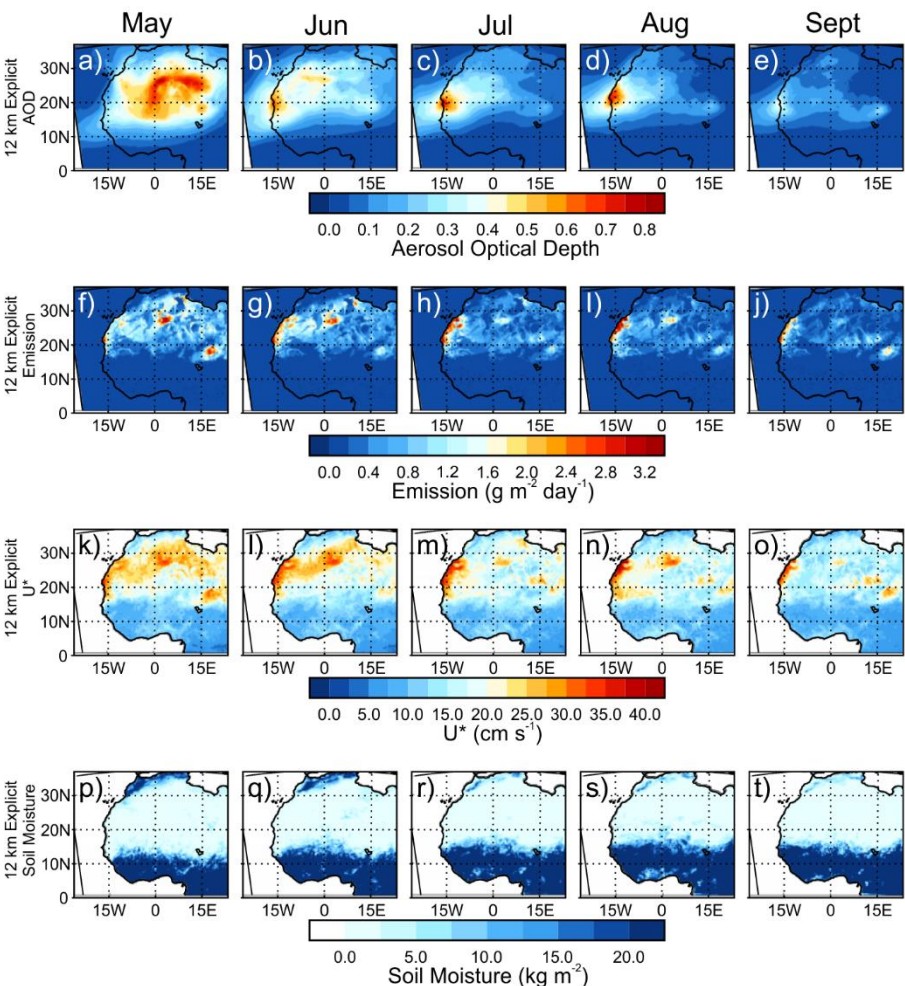

**Figure 4. Monthly mean (May-Sept) maps showing a) to e) aerosol optical depth (AOD), f) to j) dust emission, k) to o) friction velocity over bare soil (U\*) and p) to t) soil moisture from the 12 km simulation with explicit convection (12E).**





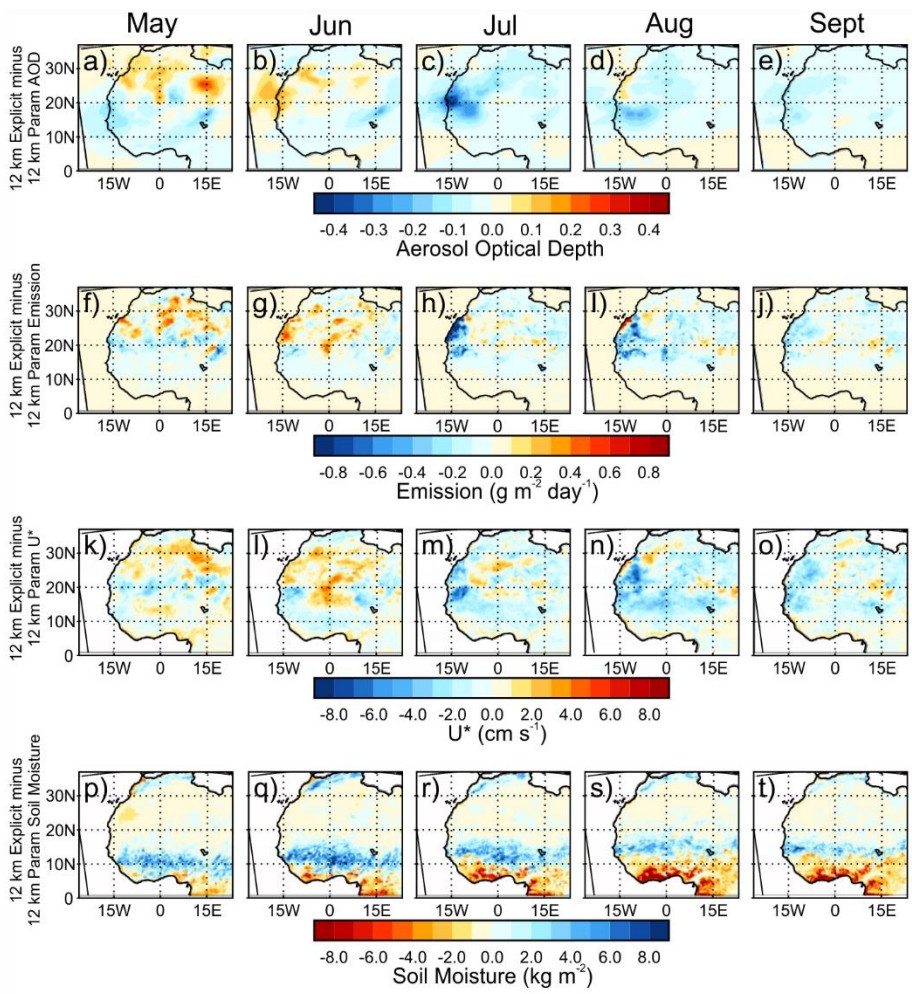

**Figure 5. Monthly means (May-Sep) difference maps showing a) to e) aerosol optical depth (AOD), f) to j) dust emission, k) to o) friction velocity over bare soil (U\*) and p) to t) soil moisture between 12km simulation with explicit convection and 12 km simulation with parameterised convection (12E – 12P).**



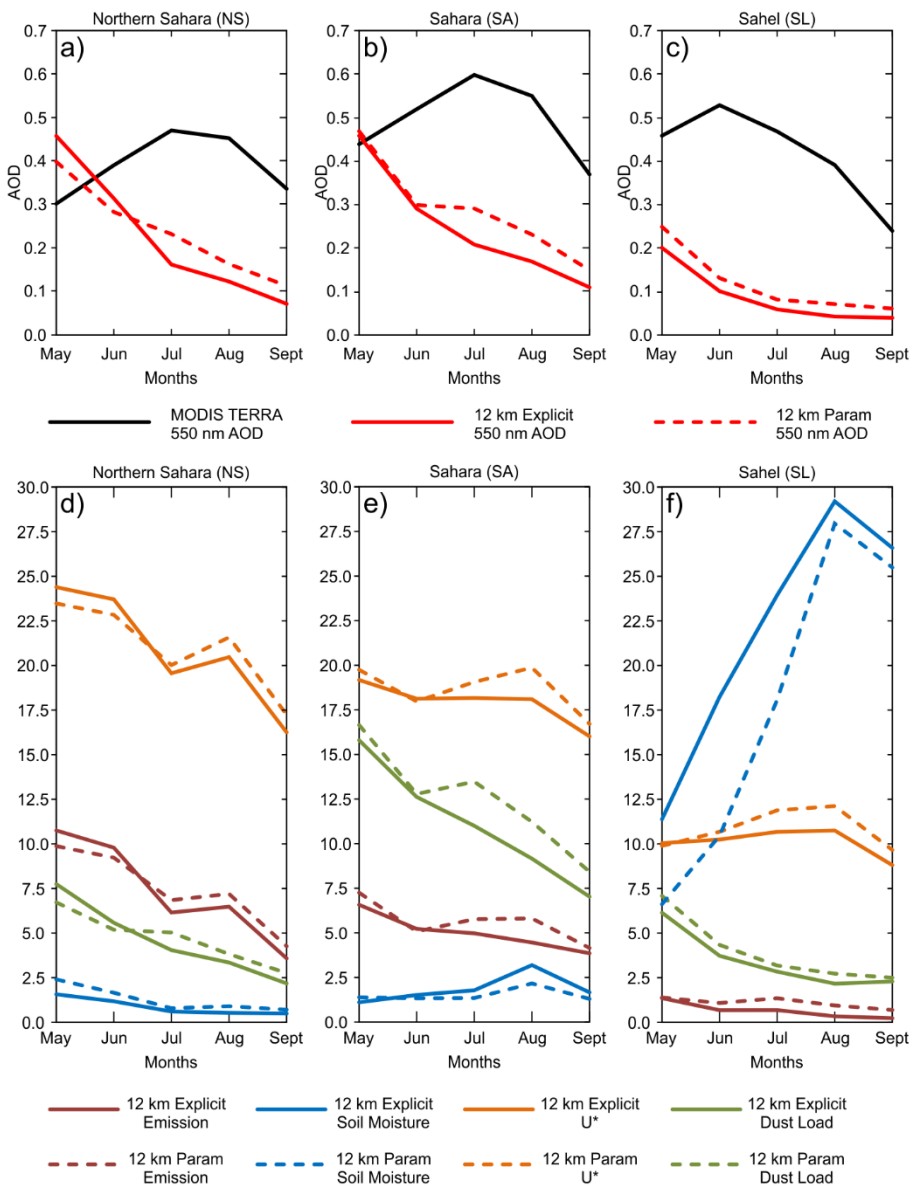

**Figure 6. Monthly mean (May-Sep) Aerosol Optical Depths averaged over a) the North Sahara (NS) region, b) the Sahara region (SA) and c) the Sahel region (SL). Shown are 12km explicit simulation (12E), 12 km parameterised simulation (12P) and MODIS (Terra). Also shown are d) NS, e) SA and f) SL dust emissions (μg m-2 s-1), total regional dust load (Tg), friction velocity (U*; cm s-1) and soil moisture in top 10cm layer (kg m-2). For ease of visibility on the plot, dust loads for NS and SA are scaled by a factor of 5 and for SL dust loads and dust emissions are scaled by a factor of 10.**

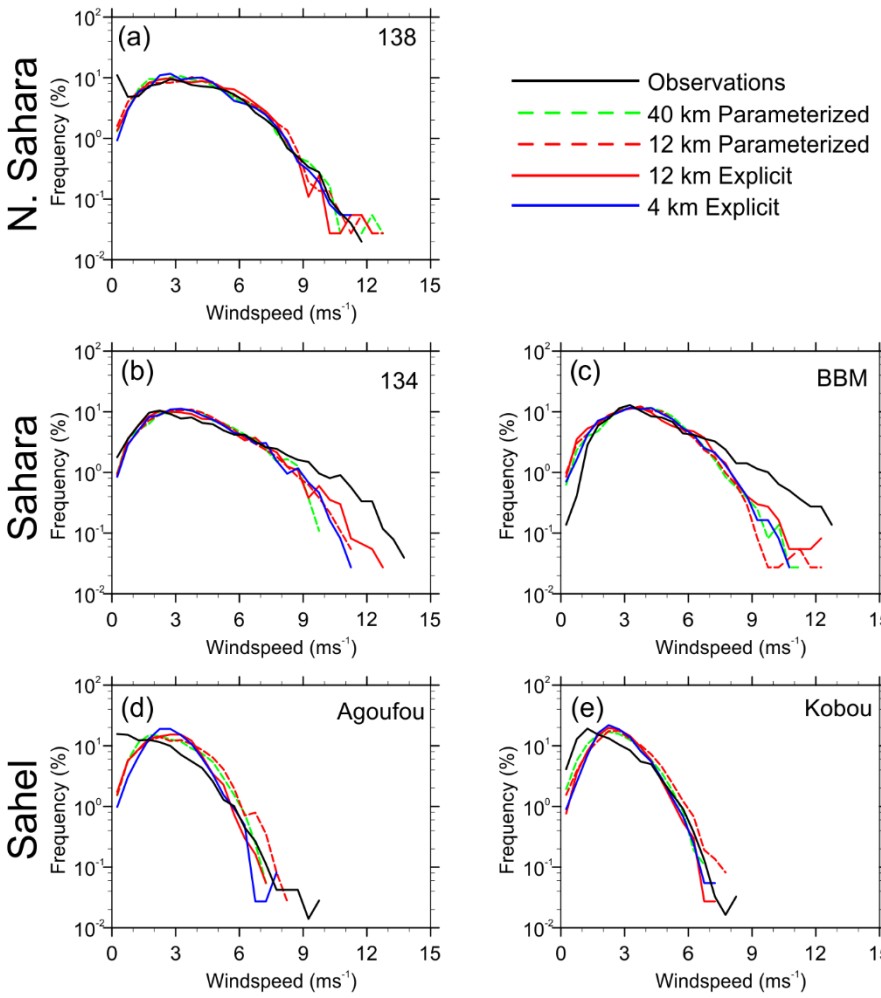

**Figure 7. Probability density functions showing the frequency of wind speeds of different strengths for the observation stations and the closest simulated grid box. Rows indicate the box from Figure 1 in which the stations are located. Black indicates observations and colours and dash pattern indicate grid-spacing and representation of convection of the 4 simulations.**




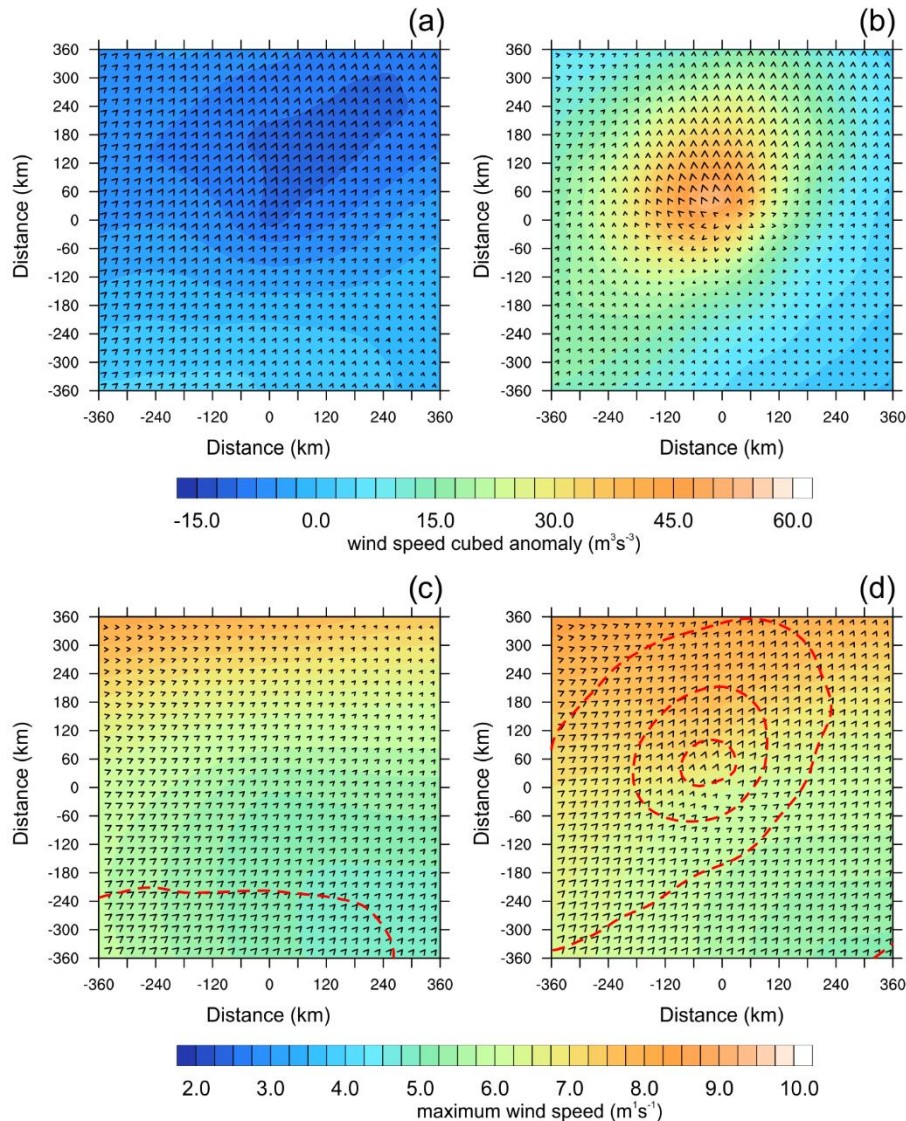

**Figure 8. Composites around column rainfall exceeding 1 mm/hr. Composited time includes the point of threshold exceedance and the following six hours of the simulation. (a) and (b) show the wind speed cubed anomaly for 12 km simulations with parameterised and explicit convection respectively, arrows represent wind anomaly. (c) and (d) show composites of the maximum wind speed in the rainfall to rainfall +6 hr window for 12 km simulations with parameterised and explicit convection respectively, arrows represent composite winds, red dashed lines represent wind speed cubed anomaly at 15 $m^3s^{-3}$ interval.**





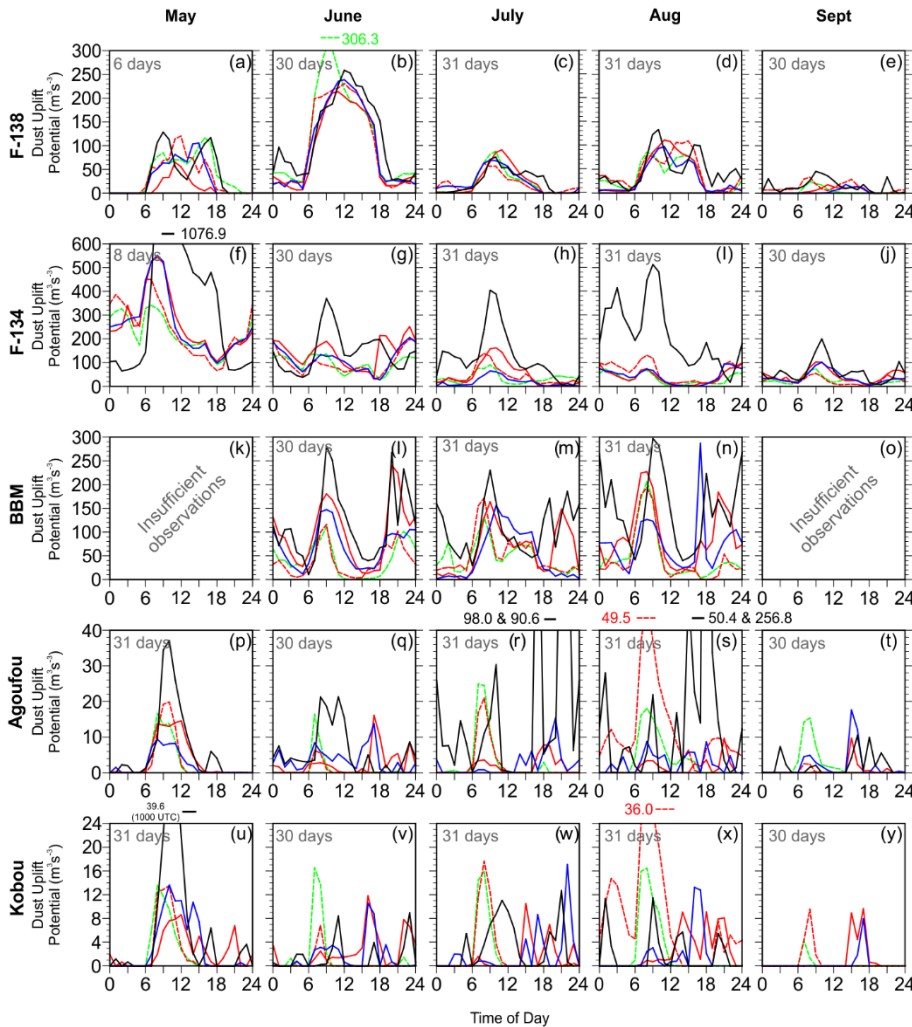

**Figure 9. Diurnal cycle of dust uplift potential at the 5 observation stations for all 5 simulated months. Colours and dash patterns same as Figure 7 (black = observations, green = 40 km simulation, red = 12 km simulations and blue = 4 km simulation, dashed lines indicate parameterisation of convection, solid lines indicate explicit convection). Where fewer than 5 days with data were available the diurnal cycle has not been calculated, for clarity the number of days with data has been shown for each panel.**





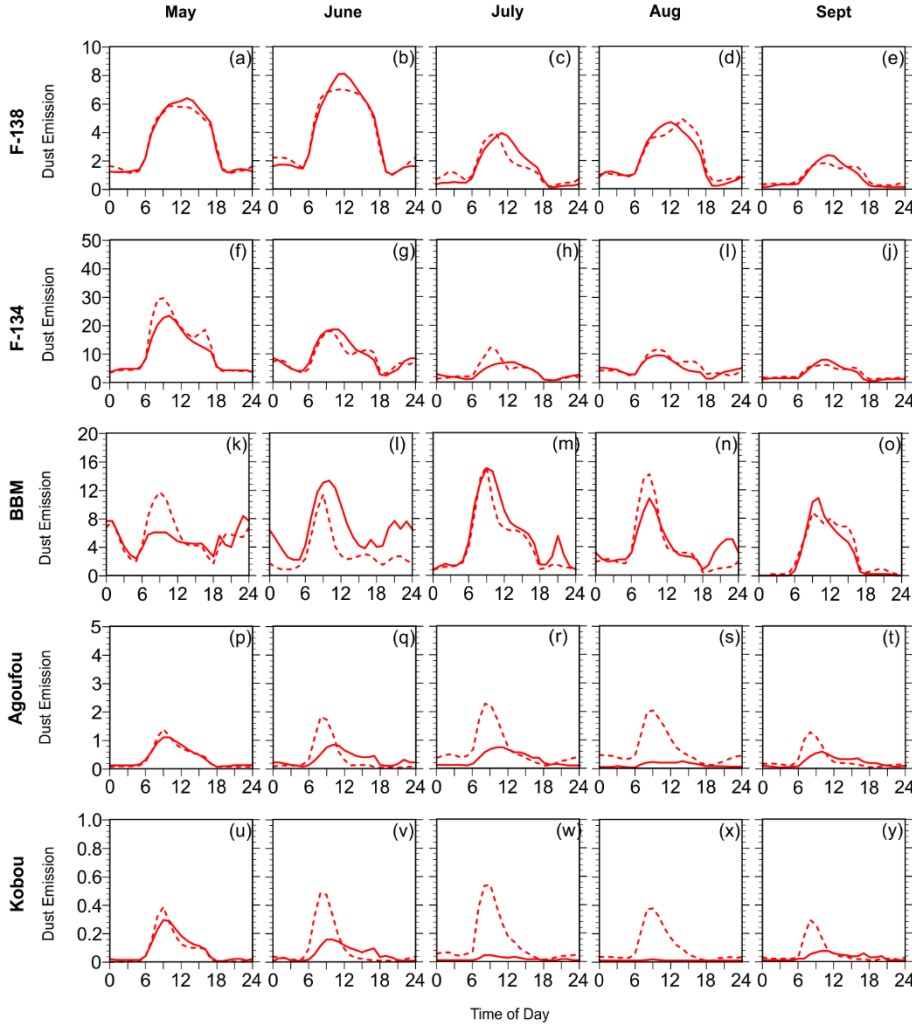

**Figure 10. Monthly mean (May-Sep) diurnal cycles in dust emission (in µg m² s⁻¹) for Fennec and AMMA stations (shown on Figure 1) for 12km models with explicit (12E, red solid line) and parameterised (12P, red dashed line) convection. a) to e) Fennec station F-138, f) to j) Fennec station F-134, k) to o) Fennec supersite at Bordj Badji Mokhtar, p) to t) AMMA CATCH site at Agoufou and u) to y) AMMA CATCH site at Kobou. Note changing vertical scale for each location.**




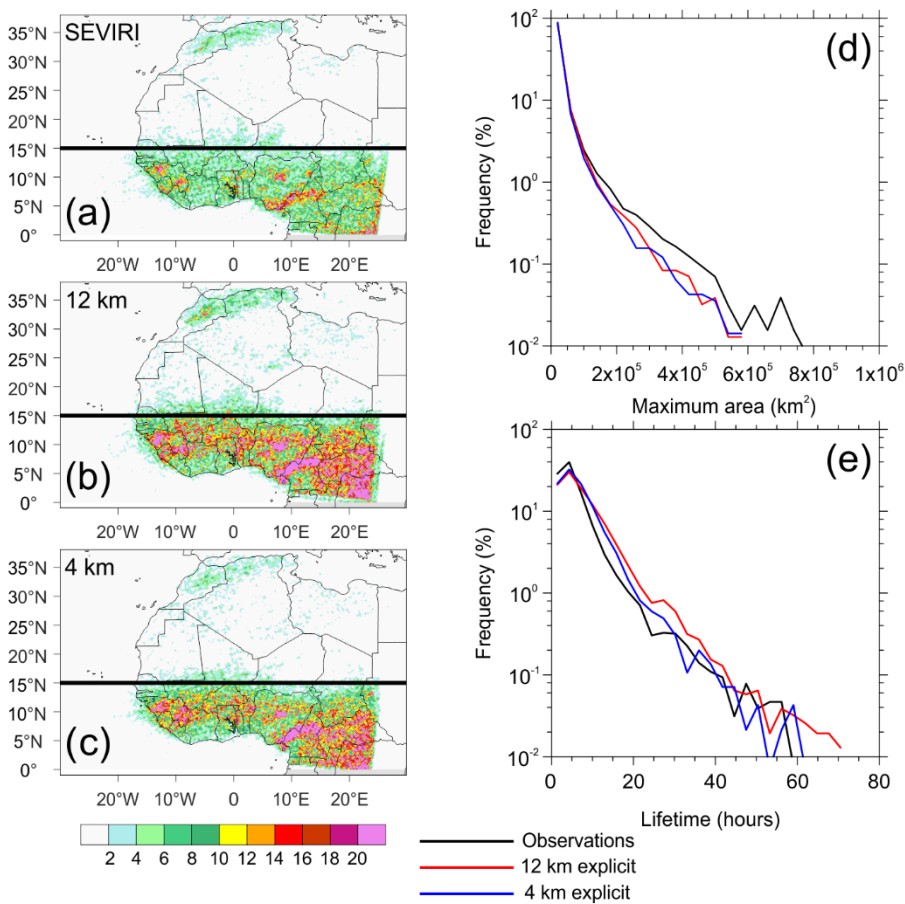

**Figure 11. Storm tracking Mesoscale Convective System (MCS) track density for (a) observed MCSs, (b) MCSs in the 12 km convection permitting simulation and (c) MCSs in the 4 km convection permitting simulation. Area and lifetime distributions for the tracked MCSs are also shown in panels (d) and (e) respectively. 15 °N is highlighted on the track density plots (a) to (c) to aid in the interpretation between tracks obtained from observations and those from simulations in the marginal region where dust is known to be raised in reality but is not raised in simulations.**



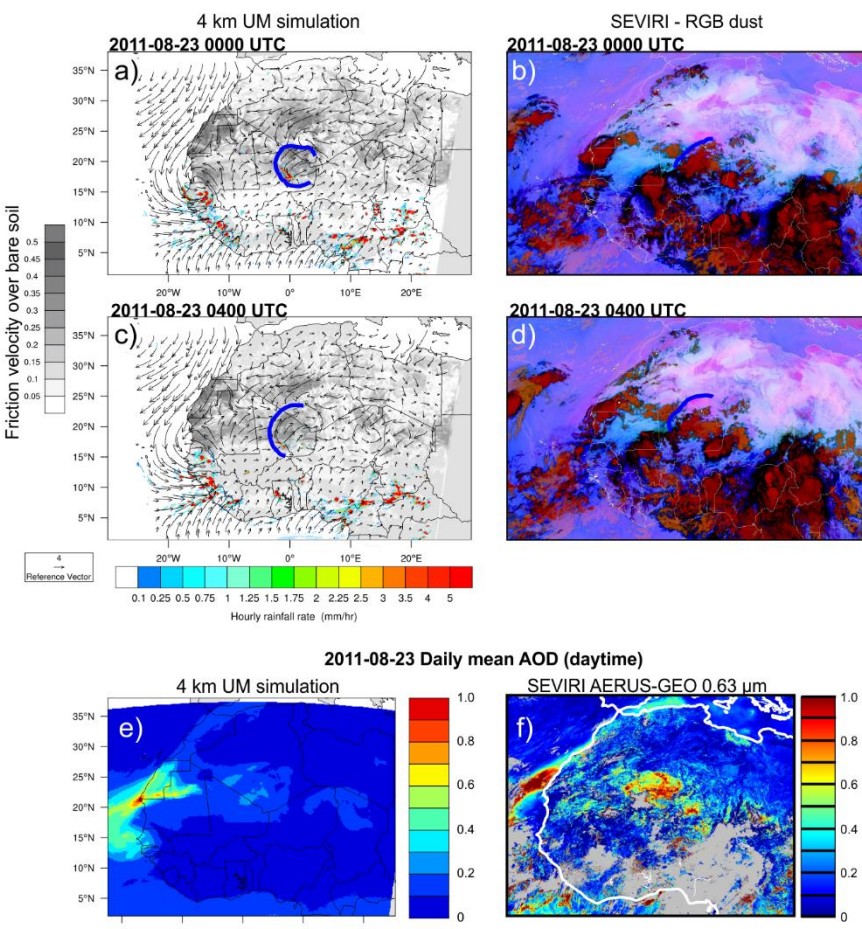

**Figure 12. Case study of a large cold pool event that occurred on the morning of 23rd August 2011 that is present both in observations and in the 4 km explicit simulation. Panels a) and c) show simulated rainfall (colours), 10 m wind (vectors) and friction velocity over bare soil (grey shading, a key feature in the emission of dust within the model). Panels b) and d) show SEVIRI false colour RGB dust imagery for the same times as panels a) and c). The leading edge of the cold pool has been highlighted in blue where visible on both plots of the simulation and SEVIRI images. Panels e) and f) show simulated and observed AOD values (daytime means) for the day of the haboob. Observed AOD is from the SEVIRI AERUS-GEO AOD product.**



| Expt name | Horiz grid len (km) | Convection type | Dust Rad Effect | NLevs | Top (km) | Tstep (min) | Subgrid Turb Mix Len Const |
|---|---|---|---|---|---|---|---|
| 4E+Fx | 4 | Explicit (3DS) | Y | 70 | 40 | 1.67 | 0.1 |
| 4E | 4 | Explicit (3DS) | N | 70 | 40 | 1.67 | 0.1 |
| | | | | | | | |
| 12E+Fx | 12 | Explicit (3DS) | Y | 70 | 80 | 2.5 | 0.05 |
| 12E | 12 | Explicit (3DS) | N | 70 | 80 | 2.5 | 0.05 |
| 12P+Fx | 12 | Parameterised | Y | 70 | 80 | 2.5 | N/A |
| 12P | 12 | Parameterised | N | 70 | 80 | 2.5 | N/A |
| | | | | | | | |
| 40P+Fx | 40 | Parameterised | Y | 70 | 80 | 2.5 | N/A |
| 40P | 40 | Parameterised | N | 70 | 80 | 2.5 | N/A |

Table 1. Summary of model simulations run in SWAMMA.