# Peer review of "Can explicit convection in the Met Office Unified Model (UM) improve modelled dust in summertime West Africa?"

_Atmospheric Chemistry and Physics, 2017_

## Referee Comment (RC1) · Anonymous Referee #1 · 30 Jan 2018

The paper focuses on the improvement in modelled dust that could be achieved when the convection is explicitly represented. A total of 8 simulations run with the UM model over a warm season (May-September 2011) are assessed in terms of dust AOD and near-surface wind speed. These simulations differ with the horizontal grid spacing (40, 12 and 4 km) and the use of a convection parameterization. Against expectation, no significant improvement in modelled dust is achieved with the convection permitting simulation. Whatever the grid spacing, the dust AOD is poorly reproduced by the UM model. The study points out two major drawbacks: a too low wind speed and a wrongly fixed bare soil parameterization. While there is no attempt to solve these issues, the study is tough of interest in the way the assessment is performed.

**Specific comments**

[Figure]

**Abstract**

Page 1, line 12, the cold pool outflows can have an important role in raising dust. However, it is not well established that their contribution can be over 50%. For example, Chaboureau et al. (2016) estimated the role of harmattan to 80% of dust emission (over the western Sahara and in June 2011).

**Introduction**

Page 2, line 30, Chaboureau et al. (2016) evaluated not only dust, but also 10-m wind speed. Even if the evaluation was done between models only (not against observations), this is of great interest as it differs much from your results (see comment below on Figure 7).

**Section 2.1**

Page 4, line 18, please give the height of the first model level. How is the calculation of the surface friction velocity $U^\star$ sensitive to the model level?

Page 5, line 16, the issue is not to make give a fair comparison. Instead, it is to simulate AOD well for the right reasons. The question is thus on the scale awareness of the dust scheme. As the dust scheme depends on the surface friction velocity, which depends itself on the grid spacing, this requires an adjustment of the model values. So does the mixing length of the 3D scheme (page 5, line 30).

**Section 3.1**

Page 10, Figure 2, does the MODIS AOD provide a reliable reference for model assessment? It should be of interest to show as well the other product you use, the SEVIRI AERUS-GEO AOD. This is valid also for Figure 6.

Page 10, Figure 3, why the models are quite good in May and very poor the other months? This seems to be due to a decrease in $U^\star$. So, why does $U^\star$ decrease with month?

**Section 3.2**

Page 12, Figure 7, Chaboureau et al. (2016) compared the 10-m wind speed over western Sahara for models with parameterized convection and without and found wind speed up to 15 m/s. This is the case for the ALADIN model run with a 24-km grid spacing for which cold pools were not expected to be simulated. This result strongly contrasts with the one shown for the UM model. This suggests a drawback in the UM model wind speed that would be not specifically related to the representation of cold pools. Further, the convection-permitting models in Chaboureau et al. (2016) show wind speed up to 25 to 30 m/s (due to cold pools), a much larger value than obtained from the UM model.
* * *

---

## Author Comment (AC1) · 8 Mar 2018

Response to Anonymous Referee #1 interactive comments on "Can explicit convection improve modelled dust in summertime West Africa?"

Reviewers comments are given in black and responses are in green.

**General Comments**

The paper focuses on the improvement in modelled dust that could be achieved when the convection is explicitly represented. A total of 8 simulations run with the UM model over a warm season (May-September 2011) are assessed in terms of dust AOD and near-surface wind speed. These simulations differ with the horizontal grid spacing (40, 12 and 4 km) and the use of a convection parameterization. Against expectation, no significant improvement in modelled dust is achieved with the convection permitting simulation. Whatever the grid spacing, the dust AOD is poorly reproduced by the UM model. The study points out two major drawbacks: a too low wind speed and a wrongly fixed bare soil parameterization. While there is no attempt to solve these issues, the study is though of interest in the way the assessment is performed.

We would like to thank the anonymous reviewer for their helpful comments on this manuscript. We agree with the reviewer that there was nearly no change in the lifting of dust in simulations despite significant changes to the simulation methodology. Instead the lack of change is investigated in this work. We conclude that changes in wind speed associated with the representation of convection are insufficient to increase the dust raising potential overall. But, even under a regime with significantly higher winds the uplift would be strongly controlled (limited) by the seasonally static bare soil fraction and the depth of the shallowest soil moisture level in the Met Office Unified Model (UM).

**Specific Comments**

**Abstract**

Page 1, line 12, the cold pool outflows can have an important role in raising dust. However, it is not well established that their contribution can be over 50%. For example, Chaboureau et al. (2016) estimated the role of harmattan to 80% of dust emission (over the western Sahara and in June 2011).

We apologise for the misunderstanding created by this statement and **have changed it to reflect the local and seasonal nature of haboob uplift.** This statement is also later qualified later in the paper, saying that "cold-pool outflows contribute over 50% of the total uplift in some areas of the Sahara in summer (Marsham et al., 2013; Allen et al., 2013 & 2014; Heinold et al., 2013)". The evidence for this claim is based upon observations of dust and wind speed in the Sahara (Marsham et al., 2013; Allen et al., 2013 & 2014) and simulations of convection permitting simulations with offline dust emission (not limited by restrictive surface characteristics; Heinold et al., 2013).

We would also like to highlight that the estimation of 80 % of emission associated with the Harmattan in Chaboureau et al., (2016) is for the "western Sahara" which excludes regions close to the Aïr and Adrar des Iforas where emission is "dominated" by afternoon emission in high resolution convection permitting simulations (haboobs). Therefore, the findings of Chaboureau et al. (2016) are not in disagreement with the fact that haboobs can (for specific regions and seasons) be responsible for over 50% of dust uplift.

We think that it is also important to recognise that simulations capable of producing organised convective storms will not automatically be able to represent the near-surface winds of real cold pools in terms of size, duration or wind strength. This creates a cold pool, dust-uplift, grey-zone where cold pools are present (allowing for convective organisation) but

are not resolved well enough to represent their full impact on dust uplift. A statement to this affect has been added for clarity.

**Introduction**

Page 2, line 30, Chaboureau et al. (2016) evaluated not only dust, but also 10-m wind speed. Even if the evaluation was done between models only (not against observations), this is of great interest as it differs much from your results (see comment below on Figure 7).

The statement indicating the difference between this work and Chaboureau et al., (2016) is to highlight the novel nature of the work and point out the differences between this study and the similar work already conducted. We agree that the difference in results is interesting and indicates that what is true for one model is not necessarily the case for another (albeit similar) model. That is why when making statements about the relevance of this work for climate modelling, we have been careful to indicate the similarity between dust fields in the SWAMMA simulations and those in CMIP5. It is also the case that work of this type will be of interest to the UM model development community given the importance of dust modelling for different stakeholders on a wide variety of timescales. We are sorry for this mistake and have clarified that the analysis in Chaboureau does include analysis of 10 m winds.

**Section 2.1**

Page 4, line 18, please give the height of the first model level. How is the calculation of the surface friction velocity U\* sensitive to the model level?

The height of the first model levels of simulations has been included for clarity (model levels are the same for the 40 and 12 km grid spaced simulations, with lowest at 10 m and are finer for the 4 km simulations, lowest level at 2.5 m). The calculation of u\* is internal to the model (boundary layer scheme) and is given as a model diagnostic.

More generally u\* can be defined as

$$u^* = \sqrt{\frac{\tau}{\rho}}$$

Where  $\tau$  is the shear stress and  $\rho$  is the air density.  $\tau$  can be defined as

$$\tau = \mu \frac{du}{dz}$$

Where du is change in wind speed, dz is height difference and  $\mu$  is the dynamic viscosity. Given the insensitivity of air density, the shear stress is the dominant term. Assuming that the lowest model level is within the turbulent boundary near the surface, (which is a reasonable assumption with lowest model levels at 10 m and 2.5 m) then u\* should be fairly insensitive to changes in the height of the lowest model level. This seems to be the case in the SWAMMA simulations with there being no clear relationship between the friction velocity values present in the 12 km simulations and those in the 4 km simulations, which have different level spacing and might have the potential to impact on friction velocity values.

Page 5, line 16, the issue is not to make give a fair comparison. Instead, it is to simulate AOD well for the right reasons. The question is thus on the scale awareness of the dust scheme. As the dust scheme depends on the surface friction velocity, which depends itself on the grid spacing, this requires an adjustment of the model values. So does the mixing length of the 3D scheme (page 5, line 30).

Wherever possible the simulations have been kept as similar to one another as is feasible. That way we can be as confident as possible that the differences in the results are due to the very small number of differences between each simulation's setup. Our results are consistent with past studies (Marsham et al 2011, Heinold et al 2013) that for the UM at these grid-spacings total dust uplift potential is almost independent of horizontal grid spacing and therefore there was no need to retune. This is in contrast to coarser grid spaced simulations (eg the Ridley et al., 2012). Allowing for a more straightforward comparison unmuddled by retuning that maybe needed for lower resolution models.

Page 10, Figure 2, does the MODIS AOD provide a reliable reference for model assessment? It should be of interest to show as well the other product you use, the SEVIRI AERUS-GEO AOD. This is valid also for Figure 6.

The MODIS combined product utilises both Dark Target and Deep Blue algorithms to be able to retrieve aerosol optical depth over both ocean and land (including arid, high albedo surfaces). Sayer et al., (2014) gives a detailed analysis of the performance of both algorithms with the MODIS AOD comparing well globally with AERONET AOD retrievals (R = 0.92) with the MODIS and AERONET AOD distribution being centred around the 1:1 line. In dusty regions there is a tendency for MODIS to underestimate AOD values. However, the spatial distribution of dust will be well represented and allow for a good comparison with the simulations compared to retrievals, the absolute value of AOD in MODIS becomes less important for analysis.

While there will always be differences between different retrieval products such as MODIS and SEVIRI AERUS-GEO these differences are small compared to the model-retrieval differences. MODIS is a much more widely used and trusted product and it is a simple procedure to download and process large amounts of data. Its use in figures 2 and 6 reflects this as regional-monthly averages can be computed. The SEVIRI AERUS product is available through the icare website (Univeristy of Lille). I had hoped that I would be able to fulfil your request to add the SEVIRI AERUS-GEO data to figures 2 and 6, however, downloading data (as opposed to imagery) for SEVIRI AERUS-GEO appears to be more complicated than anticipated. I have contacted icare but have yet to hear back about the availability of AOD data.

The SEVIRI AERUS-GEO AOD imagery remains the preferred option for figure 12 due to its better spatial coverage allowing us to show the retrieved dust distribution for a single case study event. It serves to show the absence of dust AOD associated with the MCS cold pool in the 4 km simulation and indicates that even when we might consider the simulation to have done a good job in representing deep convection, the cold pool produced is not able to raise anywhere near enough dust compared to observations.

Page 10, Figure 3, why the models are quite good in May and very poor the other months? This seems to be due to a decrease in u\*. So, why does u\* decrease with month?

In May there is a lot more dust than in other months, however their AOD maxima is too far north, it is not in a latitudinal band as seen in MODIS retrievals, and the well-known major hotspot of the Bodele depression is much too weak.

There is generally a decrease in u\* in the Sahara and Northern Sahara from May to September. However, the pattern in the Sahel is less clear as shown in Figure 6. Given that emission is a cubic function of u\* associated with exceedance of a u\* threshold, it is no surprise that with dropping u\* values (possibly associated with a weakening Harmattan after the start of the monsoon onset and developing Saharan Heat Low) that AOD values would also fall. We see that there is only a weak growth in afternoon and evening winds and an unrealistic drop in nocturnal low level jet winds in convection permitting simulations. As discussed in the paper, this can be attributed to cold pools that are too small and too weak. Cold pools are known to be key to dust uplift in the Sahel and southern Sahara from observations (Marsham etal., 2013, Allen et al., 2013, 2014) and given the recirculation of dust around the Saharan Heat Low would go a long way to explaining the peak dust location growing through June, July and August in MODIS AOD.

Page 12, Figure 7, Chaboureau et al. (2016) compared the 10-m wind speed over western Sahara for models with parameterized convection and without and found wind speed up to 15 m/s. This is the case for the ALADIN model run with a 24-km grid spacing for which cold pools were not expected to be simulated. This result strongly contrasts with the one shown for the UM model. This suggests a drawback in the UM model wind speed that would be not specifically related to the representation of cold pools. Further, the convection-permitting models in Chaboureau et al. (2016) show wind speed up to 25 to 30 m/s (due to cold pools), a much larger value than obtained from the UM model.

The model wind-speeds plotted in figure 7 have been height adjusted for comparison against observed winds at 2 m. This means that the 10 m model winds will be higher than those shown in Figure 7. We have added information about this technique and indicated that this brings modelled winds speeds close to those simulated in parameterised models in Chaboureau et al., (2016).

The indication that the convection permitting models in Chaboureau et al., (2016) have an increase in frequency of strong winds associated with haboobs does suggest that they are doing a better job that the UM in representing the effects of cold pools. The work presented here indicates that (for the UM at least) that the cold pools produced are too small, too weak and do not raise dust in a way that is consistent with in situ observations or satellite retrievals, it is also possible that the increased windiness shown in Chaboureau et al. (2016) is also too weak. We understand that this result does contrast with that of Chaboureau et al., (2016) and have added some detail about the differences. We have also indicated that the conclusions made are primarily relevant for the UM and simulations that share similarly poor AOD distributions, such as those in CMIP5.

---

## Referee Comment (RC2) · Anonymous Referee #2 · 23 Mar 2018

The paper's goal is to understand whether including explicit convection improves the modeled dust in summertime West Africa. In the model used, there is an improved diurnal cycle, but the average dust aerosol optical depth is only slightly modified with explicit convection because the increased evening dust is balanced by a reduction of morning dust (associated with the breakdown of the low-level jet). The results show increases in the frequency of the strongest winds but they are still weaker than observed. Finally, the authors conclude that their study is limited due to other model problems such as the poor representation of the land surface condition in the Sahel, where haboobs are frequently generated in summer.

Although some of the results of the study are interesting, I have some concerns with respect to the formulation of the research question, the experimental set up and the

interpretation of the results. I summarize these concerns below.

1) Research question: In my opinion, the content of the paper cannot respond to the question posed in the title: "Can explicit convection improve modelled dust in summertime West Africa? What the study shows is that errors from neglecting explicit convection are of second order compared to other model errors, and the conclusion can only apply to the limited area version of the UM of the UK Met Office used in the paper. There may be other models with a similar behavior but this is not shown in the paper, and extrapolating would be speculative. It would be convenient to modify the title, otherwise it can be confusing for the reader (in terms of what the reader expects by reading the title). The same happens in the abstract and the paper: it has to be clear that these results are specific to the UM.

2) Figures 2 and 6 show a very poor behavior of the model (regardless of explicit convection) when compared to observations. For example, there is a factor 3 to 4 difference in the AOD and an uncorrelated seasonality when comparing the model to the MODIS observations. I have several questions in this respect:

a. To what extent the retrieved AOD form MODIS is reliable over land? Some measure of uncertainty in the observation is needed when using AOD products over bright surfaces.

b. Have the model outputs been spatially and temporally collocated with the MODIS data in order to perform the comparison (i.e. did you select the modeled times and places corresponding to the availability of the MODIS data?) If not, to what extent your comparison can be affected? My experience is that it matters a lot. Would this make sense with your current model set up, i.e. a regional climate run only fed by the analysis data though the boundaries?

c. Why AERONET stations were not used? There are quite a few stations in the domain for 2011. AERONET is reliable and is the main tool used to evaluate model performance. Without the AERONET evaluation is difficult to judge the performance of
this model compared to other models. Nowadays many regional models represent reasonably well the seasonality of dust in AERONET stations (daily correlations between 0.6 and 0.8 when reinitialized daily and without dust data assimilation). There are also available high resolution PM10 surface observation concentrations for the Sahelian Transect (Marticorena et al 2010) that would really help evaluating the model.

d. Concerning the previous point: in the introduction the authors claim that both winds and dust should be explored together with observations. It is surprising that the authors do not use the most reliable resources of dust measurements besides the more uncertain satellite products.

e. Concerning the general decrease of dust in the model from May to September (compared to the observations showing a peak around July): Given that the model is not reinitialized every, has the humidity in May 1 been warmed up for at least 1 year? If not, this could be a reason for such behavior (the model could be showing a trend in dust because of a drift in the soil humidity). Has the model been evaluated for the same time period reinitializing the atmosphere and the soil every day from the parent domains?

f. More details should be given on the emission scheme. Do the authors use a preferential source? Do they use estimates of aerodynamic roughness length? This may also at least partly explain such a mismatch with observations.

3) A major question: because convection seems to be a second order error in this study, can we really respond to the question posed in the title?

4) What can explain that the 12 Km explicit and 4 km explicit are so similar?

5) In Figure 7, the 12-km explicit has a more prominent tail of high winds compared to the 4-km explicit. This behavior is surprising to me. What does explain this behavior? Is the frequency at a specific location comparable using different model resolutions? That is not really clear to me.

[Figure]

6) Figure 12: Does it make sense to compare the model for a specific day for this experimental set up? Reproducing a specific episode requires (recent) and accurate initial conditions and the model is running in a regional climate mode only constrained through the boundaries.

---

## Author Comment (AC2) · 4 Apr 2018

Response to Anonymous Referee #2 interactive comments on "Can Explicit convection improve modelled dust in summertime West Africa?"

Reviewers comments are given in black and responses are in green.

The paper's goal is to understand whether including explicit convection improves the modeled dust in summertime West Africa. In the model used, there is an improved diurnal cycle, but the average dust aerosol optical depth is only slightly modified with explicit convection because the increased evening dust is balanced by a reduction of morning dust (associated with the breakdown of the low-level jet). The results show increases in the frequency of the strongest winds but they are still weaker than observed. Finally, the authors conclude that their study is limited due to other model problems such as the poor representation of the land surface condition in the Sahel, where haboobs are frequently generated in summer.

Although some of the results of the study are interesting, I have some concerns with respect to the formulation of the research question, the experimental set up and the interpretation of the results. I summarize these concerns below.

1) Research question: In my opinion, the content of the paper cannot respond to the question posed in the title: "Can explicit convection improve modelled dust in summertime West Africa?" What the study shows is that errors from neglecting explicit convection are of second order compared to other model errors, and the conclusion can only apply to the limited area version of the UM of the UK Met Office used in the paper. There may be other models with a similar behavior but this is not shown in the paper, and extrapolating would be speculative. It would be convenient to modify the title, otherwise it can be confusing for the reader (in terms of what the reader expects by reading the title). The same happens in the abstract and the paper: it has to be clear that these results are specific to the UM.

Firstly, we would like to thank the reviewer for their time and the helpful comments they have made on this manuscript.

The title of the paper is meant to indicate that the expectation we had on starting this work. We had thought that by representing convection explicitly we would see a marked improvement in the dust aerosol fields when comparing with simulations with parameterised convection and satellite retrievals of Aerosol Optical Depth (AOD). We tried to make this clear in the "story" of the paper with section 3.1 outlining the lack of effect of representation of convection on the overall AOD and the subsequent results sections investigating the differences in near surface wind and modelled storms. In that respect we don't believe the title to be misleading, the answer to the posed question is "no", and we have then attempted to explain the reason that this is the case.

However, we are also aware that the results shown in this work are limited to a single setup of the Met Office Unified Model (UM) running in a limited area setup without reinitialisations. It is plausible that simulations performed using models from different centres would produce different responses to a change in the representation of convection. **Therefore, we have modified the title of the work to indicate that this work focusses on the UM in a limited area framework. We have also endeavoured to make sure that in the discussion of**

**results throughout the paper that it is made clear that other models might respond differently.**

2) Figures 2 and 6 show a very poor behavior of the model (regardless of explicit convection) when compared to observations. For example, there is a factor 3 to 4 difference in the AOD and an uncorrelated seasonality when comparing the model to the MODIS observations. I have several questions in this respect:

a. To what extent the retrieved AOD form MODIS is reliable over land? Some measure of uncertainty in the observation is needed when using AOD products over bright surfaces.

Over bright surfaces the combined product relies on the Deep Blue algorithm. The implementation of the Deep Blue algorithm has provided a much improved technique for the retrieval of AOD values over bright surfaces compared to the Dark Target algorithm. By using maps and libraries of surface reflectance in the blue part of the spectrum the Deep Blue algorithm is able to retrieve AOD values that compare well with AERONET measurements. The technique is described in detail in Hsu et al., (2013) and has an estimated error better than 0.05 + 20 %, with 79 % of the best AOD data falling within this range. **Detail about the accuracy of the product have now been included in the manuscript.**

This concern has also been addressed in the comments to reviewer 1 (see below).

"The MODIS combined product utilises both Dark Target and Deep Blue algorithms to be able to retrieve aerosol optical depth over both ocean and land (including arid, high albedo surfaces). Sayer et al., (2014) gives a detailed analysis of the performance of both algorithms with the MODIS AOD comparing well globally with AERONET AOD retrievals (R= 0.92) with the MODIS and AERONET AOD distribution being centred around the 1:1 line.

In dusty regions there is a tendency for MODIS to underestimate AOD values. However, the spatial distribution of dust will be well represented and allow for a good comparison with the spatial distribution of modelled dust. Also, given the very low values of AOD in the simulations compared to retrievals, the absolute value of AOD in MODIS becomes less important for analysis."

b. Have the model outputs been spatially and temporally collocated with the MODIS data in order to perform the comparison (i.e. did you select the modeled times and places corresponding to the availability of the MODIS data?) If not, to what extent your comparison can be affected? My experience is that it matters a lot. Would this make sense with your current model set up, i.e. a regional climate run only fed by the analysis data though the boundaries?

We would like to apologise for not making this clear. In all comparisons between MODIS and models, simulations are sub sampled temporally to limit the potential errors introduced by the diurnal variations and spatially to account for missing MODIS data (due to clouds). The comparison of mapped monthly averages and, box averages also reduces the introduction of erroneous analysis that would result from using a single model grid box or observation site and interpolating a pattern across a wider region. **The manuscript has been updated to make this treatment of model data clearer.**

c. Why AERONET stations were not used? There are quite a few stations in the domain for 2011. AERONET is reliable and is the main tool used to evaluate model performance. Without the AERONET evaluation is difficult to judge the performance of this model

compared to other models. Nowadays many regional models represent reasonably well the seasonality of dust in AERONET stations (daily correlations between 0.6 and 0.8 when reinitialized daily and without dust data assimilation). There are also available high resolution PM10 surface observation concentrations for the Sahelian Transect (Marticorena et al 2010) that would really help evaluating the model.

Even without AERONET or surface PM10 measurements we are able to see that, with respect to dust, the simulations are doing a poor job. The location where dust is being raised and where it reaches its greatest AOD values do not match well with satellite observations, therefore we didn't feel it was necessary to include further (single point) observations of dust. Were the simulations doing a better job of replicating the spatial and temporal variations of dust we think that AERONET or the Sahelian transect would have been useful in further validating the simulations, however, given the large simulation/retrieval discrepancies already shown we cannot see how this could add extra insight.

Operationally, dust models will be able to be extensively tuned for overall dust loading to match observed values, that is not the case here. One of the reasons for the experimental design was to have conditions similar to a true, free running simulation and allow insight into the kind of behaviour that is produced by the model in climate simulation conditions. It is imperative that we try to understand the factors that dominate dust uplift in models constrained in this way and try to understand the limitations of simulations in representing real physical processes. **This sentiment has been strengthened in the paper.**

d. Concerning the previous point: in the introduction the authors claim that both winds and dust should be explored together with observations. It is surprising that the authors do not use the most reliable resources of dust measurements besides the more uncertain satellite products.

It was felt that there was a greater value in having a product with a broader spatial coverage, that allowed the regional pattern of atmospheric dust to be compared with simulated dust fields. As mentioned above, had the simulations shown a better spatial distribution of dust then it is likely that other observations (for model validation) would have been required.

e. Concerning the general decrease of dust in the model from May to September (compared to the observations showing a peak around July): Given that the model is not reinitialized every, has the humidity in May 1 been warmed up for at least 1 year? If not, this could be a reason for such behavior (the model could be showing a trend in dust because of a drift in the soil humidity). Has the model been evaluated for the same time period reinitializing the atmosphere and the soil every day from the parent domains?

We believe that the month on month reduction in AOD can most convincingly be explained by the reduction in u* values (especially in the Sahara and Northern Sahara boxes). This has been discussed in the reply to reviewer #1 (see below).

"There is generally a decrease in u* in the Sahara and Northern Sahara from May to September. However, the pattern in the Sahel is less clear as shown in Figure 6. Given that emission is a cubic function of u* associated with exceedance of a u* threshold, it is no surprise that with dropping u* values (possibly associated with a weakening Harmattan after the start of the monsoon onset and developing Saharan Heat Low) that AOD values would also fall. We see that there is only a weak growth in afternoon and evening winds and an unrealistic drop in nocturnal low level jet winds in convection permitting simulations. As discussed in the paper, this can be attributed to cold pools that are too small and too weak. Cold pools are known to be key to dust uplift in the Sahel and southern Sahara from observations (Marsham etal., 2013, Allen et al., 2013, 2014) and given the recirculation of

dust around the Saharan Heat Low would go a long way to explaining the peak dust location growing through June, July and August in MODIS AOD."

The greatest control on soil moisture in the simulations is rainfall. The Northern Sahara box shows drying throughout the modelled period (making uplift more likely), the Sahara box shows little change and there is a large increase in the Sahel (associated with the monsoon rains; all shown on figure 6). We think that this kind of behaviour is to be expected and doesn't constitute a "drift" in soil humidity that would reduce uplift across the region over the simulated period.

f. More details should be given on the emission scheme. Do the authors use a preferential source? Do they use estimates of aerodynamic roughness length? This may also at least partly explain such a mismatch with observations.

The dust emission scheme is that with the Met Office Coupled Large-scale Atmosphere Simulator for Studies in Climate (CLASSIC) and is described in detail in Johnson et al., (2011). Dust emission is calculated at each time step using prognostic model fields. The widely used approach of Marticorena and Bergametti (1995) is employed where the horizontal flux of sediment in 9 bins is calculated (bin sizes are 0.0316, 0.1, 1.0, 3.16, 10.0, 31.6, 100.0, 316.0, 1000.0 µm radius). This is given by

$$G_i = \rho B U^{*3} \left( \frac{1 + U_{ti}^*}{U^*} \right) \left( 1 - \left( \frac{U_{ti}^*}{U^*} \right)^2 \right) M_i \frac{CD}{g}$$

Where i refers to bin number, G is the horizontal flux, $\rho$ is the air density at the surface, B is the bare soil fraction, $U^*$ is the friction velocity over bare soil, $U_t^*$ is the threshold friction velocity, M is the mass fraction of soil particles in the bin, C is the constant of proportionality, D is a tunable parameter and g is acceleration due to gravity. Emission is also inhibited if snow is present, the ground is frozen, on steep slopes, if soil moisture is too high and at costal grid points with fractional land cover.

The $U^*$ value used for emission calculations is generated in the Joint UK Land Environment simulator (JULES) scheme. $U^*$ is largely a function of changing wind speed with height. Therefore the surface roughness (but not orographic roughness e.g. vegetation) is an important factor in the calculation of dust emission. M is dependent on the soil characteristics in a particular grid box (from ancillary files) and takes into consideration soil clay, silt and sand fractions. The fraction of clay in a particular soil also impacts how soils respond to moisture by modifying the threshold friction velocity depending on clay fraction and soil moisture according to the method of Fécan et al., (1999).

The vertical transport of dust away from the surface is linked to the horizontal sediment transport by

$$F = 10^{(13.4F_c - 8.0)} \sum_{i=1}^{9} G_i$$

Where F is the vertical flux of dust away from the surface, Fc is clay fraction. Emission and transport away from the surface only takes place in the smallest 6 size bins.

As such, there is no explicit preferential dust sources used, however, where soil characteristics and surface roughness conditions are favourable, the threshold $U^*$ value for emission can be reduced, allowing for favoured emission in particular regions. This could go

some way to explaining erroneously high-uplift regions such as that along the Atlantic coast, however, this would not explain the near zero emission in the Sahel. **Statements to this effect have been added to the paper to clarify.**

3) A major question: because convection seems to be a second order error in this study, can we really respond to the question posed in the title?

While we appreciate that assessing the impact of convection permitting simulations on dust uplift cannot be fully investigated, we believe that the initial question is still valid. The answer to the question "Can Explicit convection improve modelled dust in summertime West Africa?" is "not currently" (at least in the UM). we think that is a well understood problem that parameterised convection in models limits their ability to represent the strongest winds for important dust uplift regions. However, just because the results from this work were unexpected should not be a reason to alter the initial question.

4) What can explain that the 12 Km explicit and 4 km explicit are so similar?

We would not expect there to be a large difference between the 12 km and 4 km explicit simulations. Previous work as part of the Cascade project (Birch et al., 2014) showed that the largest differences in this region were the result of changing the representation of convection, not modifying grid-scale. Over West Africa it seems that the scale of the convective storms (hundreds of km) means that they can be represented in simulations that would generally be considered to have too coarse a grid-spacing for the representation of convection. There are key differences between different explicit simulations and observed storms, some of which are likely to be the result of simulation grid-scale, however, those results are beyond the scope of this paper.

5) In Figure 7, the 12-km explicit has a more prominent tail of high winds compared to the 4-km explicit. This behavior is surprising to me. What does explain this behavior? Is the frequency at a specific location comparable using different model resolutions? That is not really clear to me.

There are obviously likely to be some differences due to the differing grid-box scales between different simulations, however, the fact that all the simulations (from 40 km to 4 km) show remarkably similar wind speed distributions suggests that the grid-scale effect is relatively small. I'm not sure that the differences between the 12 km and 4km explicit simulations at the high wind speeds tail are so great that we would consider them to be an important feature. In fact the only clear differences that we think are important for high wind speeds is the shortfall in frequency of all simulations when compared to observed winds at the Saharan stations.

6) Figure 12: Does it make sense to compare the model for a specific day for this experimental set up? Reproducing a specific episode requires (recent) and accurate initial conditions and the model is running in a regional climate mode only constrained through the boundaries.

The aim of this figure was to show several behaviours in the simulations: (1) that simulations with explicit convection are capable to generating large MCS storms, (2) the large MCSs (when present in simulations) are capable of generating convective cold pools that spread into the Sahara, (3) that when cold pools are present they do produce a maxima in wind speed (or friction velocity) that we would expect to be associated with the raising of dust and

(4) that the emission of dust from such an event is lower than might be expected when compared to similar observed events.

In order to highlight these points the storms chosen from observations and simulations did not have to be on the same day or time. However, it so happened that the storm generated on 23rd August 2011 in the 4 km simulation was a good match for an actual storm that occurred on that date. This suggests that information from the lateral boundary conditions (hourly global UM simulation reinitialised every 6 hours using ERA-Interim data) was able to produce conditions favourable to convection near the middle of the model domain, despite being such a long time into the simulation.

Even if these events had been on different days it would still be possible to show that points 1 to 4 mentioned above are all true. The weak uplift from event in the simulation highlights a real issue in the simulation of dust emission. However, it is important to note that whether the cause is weak winds or a problem of surface characteristics is beyond the remit of this study.

---

## Author Response (AR2)

**Co-Editor Decision: Publish subject to minor revisions (review by editor) (02 May 2018) by Yves Balkanski**

Comments to the Author:

This paper could have more impact if the remarks from the reviewers were more fully taken into 5 consideration. I would like the authors to go back to the following three comments from reviewers and insert in the text several sentences to make the best attempt to account for them:

We apologise for where we may have missed any points requested in the original reviews, and would like to thank the co-editor for taking the time to consider the reviews in detail. We have included additional sentences in the manuscript to address the three issues raised below. Details are in our replies to each point below.

- 1- Figures 2 and 6 show a very poor behavior of the model (regardless of explicit convection) when compared to observations. For example, there is a factor 3 to 4 difference in the AOD and an uncorrelated seasonality when comparing the model to the MODIS observations. I have several questions in this respect
- 15 Have the model outputs been spatially and temporally collocated with the MODIS data in order to perform the comparison (i.e. did you select the modeled times and places corresponding to the availability of the MODIS data?) If not, to what extent your comparison can be affected? My experience is that it matters a lot. Would this make sense with your current model set up, i.e. a regional climate run only fed by the analysis data though the boundaries?

**20**

Apologies that this was not made clearer. Yes, for Figures 2 and 3 MODIS and simulation comparisons are made using monthly mean values with the simulations sub-sampled to the appropriate time of MODIS overpass. This allows for fair comparison with observations across the model domain gives reliable temporal cover across the simulation period, allowing for comparison

- of monthly values. Figure 6 uses monthly mean values calculated using all available simulation data. This is so overall AOD values can be compared with the other factors that impact on dust uplift in the simulations included on Figure 6 such as U\* and soil moisture. In the context of Figure 6 MODIS AOD monthly means are included to show the observed trends in AOD (rather than absolute AOD values which would be impacted by the diurnal cycle).
- 30 Given the model setup and the paper objectives it was important to evaluate the overall spatial AOD pattern on a monthly timescale. Part of the utility of this work is to inform model development

for the better simulation of the earth-atmosphere system. Both for short term predictions and for future climate simulations. Given the poor behaviour of the simulations with respect to AOD (and the noted similarity of its behaviour to CMIP5 simulations) it is important to understand what model errors produce such issues.

5 The sentences below have been added for clarity:

"Where appropriate simulations are similarly sub-sampled to the approximate MODIS TERRA overpass time to reduce erroneous comparisons of different parts of the diurnal cycle. This gives a good spatial comparison of AOD on the monthly timescales that are studied in this work."

"As Figure 2 shows monthly mean values of AOD at approximately 1000 UTC for both simulations and satelliteretrievals we believe that this is a good comparison to judge the overall differences in the spatial distribution of dust in both reality and the simulations."

"Figure 6 summarises the seasonal trends of AOD and in factors affecting the dust AOD in the 12E and 12P models for the North Sahara (NS), Sahara (SA) and Sahel (SL) regions. We see that for both simulations the AODs (monthly mean values of all available times) are poorly simulated with their highest values in May, whereas MODIS AOD

- 15 increases from May to a maximum in July (for NS and SA) and June (for SL). The trend in AOD shown by MODIS retrievals is consistent with the summertime northwards advance of the monsoon, rainfall and haboobs (Marsham et al., 2008)."
- 2- On the following comment, you should acknowledge that a comparison to local sites and notonly for AOD but also for surface concentrations and deposition would help you understand the source of the differences between model and observations:

"Why AERONET stations were not used? There are quite a few stations in the domain for 2011. AERONET is reliable and is the main tool used to evaluate model performance. Without the AERONET evaluation is difficult to judge the performance of this model compared to other

- 25 models. Nowadays many regional models represent reasonably well the seasonality of dust in AERONET stations (daily correlations between 0.6 and 0.8 when reinitialized daily and without dust data assimilation). There are also available high resolution PM10 surface observation concentrations for the Sahelian Transect (Marticorena et al 2010) that would really help evaluating the model."
- 30 We apologise for not sufficiently addressing this issue from the reviews and have now modified the paper to address this point.

In the process of performing the work for this paper comparisons with AERONET data were performed (see Figure A below, locations in Figure B). Figures A and B show significant problems with the spatial and temporal coverage. In particular, some months/locations had incomplete data introducing significant sampling biases when comparing retrievals and simulations (most

- 5 obviously at Bordj Badji Mokhtar; one of only 2 stations in the central Sahara). AERONET stations tend to be located outside of the central Saharan region. This means that across longer simulation periods (which encompass seasonal variations) models can produce reasonable correlations due to accurate dust transport. (e.g. Pope et al., 2016 doi: 10.1002/2016GL070621 shows a correlation of 0.61 to 0.82 for AODs from the operational Unified Model with no data assimilation
- 10 with AERONET, despite missing the haboobs that generate much of the dust in reality). Here simulations are for the NH summer only, when even in analysed model products, winds have low skill compared to observations (Roberts et al., 2017 doi: 10.1002/asl.765). As shown in Figure A correlation coefficients between AERONET and simulations in the central Sahara (Tamanrasset) are very low (between 0.26 and 0.33) with higher values in most of the AERONET stations
- 15 surrounding the Sahara. This is because the transport of dust (driven by large scale wind patterns) is a much easier aspect of modelling than accurate dust emission which is reliant on correctly representing a number of effects such as high wind speed tails of wind speed distributions, surface cover, soil moisture and surface roughness.

The SWAMMA simulations present dust fields that are very similar to those seen in other

- 20 simulations (CMIP5). This pattern is not changed by the explicit representation of convection, which introduces emission from haboobs, which are known to dominate dust emission in parts of the summertime Sahel and central Sahara. The MODIS comparison shows this most clearly. While surface PM10 comparisons (and AERONET) would be interesting, the key point of the modelled dust evaluation required for this paper, is to identify major errors in the spatial and
- 25 seasonal evolution of the dust fields in the model when compared with observations. Doubtless, future studies on details of Sahelian emission would benefit from PM10 evaluation as you propose, and as we now note in the text, but that is not needed to address the aims of this paper. It is also the case that the AMMA dust transect stations are located in regions in which we know the model emission is doing a very poor job due to the inappropriate treatment of surface
- 30 characteristics (vegetation and soil moisture).